# Highly localized intracellular Ca$^{2+}$ signals promote optimal salivary gland fluid secretion

**Takahiro Takano[1], Amanda M Wahl[1], Kai-Ting Huang[1], Takanori Narita[2], John Rugis[3], James Sneyd[3], David I Yule[1]\***

[1]Department of Pharmacology and Physiology, University of Rochester, Rochester, United States; [2]Department of Veterinary Medicine, College of Bioresource Sciences, Nihon University, Fujisawa, Japan; [3]Department of Mathematics, University of Auckland, Auckland, New Zealand

**Abstract** Salivary fluid secretion involves an intricate choreography of membrane transporters to result in the trans-epithelial movement of NaCl and water into the acinus lumen. Current models are largely based on experimental observations in enzymatically isolated cells where the Ca$^{2+}$ signal invariably propagates globally and thus appears ideally suited to activate spatially separated Cl and K channels, present on the apical and basolateral plasma membrane, respectively. We monitored Ca$^{2+}$ signals and salivary secretion in live mice expressing GCamp6F, following stimulation of the nerves innervating the submandibular gland. Consistent with in vitro studies, Ca$^{2+}$ signals were initiated in the apical endoplasmic reticulum. In marked contrast to in vitro data, highly localized trains of Ca$^{2+}$ transients that failed to fully propagate from the apical region were observed. Following stimuli optimum for secretion, large apical-basal gradients were elicited. A new mathematical model, incorporating these data was constructed to probe how salivary secretion can be optimally stimulated by apical Ca$^{2+}$ signals.

**\*For correspondence:**
david_yule@urmc.rochester.edu

**Competing interests:** The authors declare that no competing interests exist.

## Introduction

Salivary fluid secretion is vital to the health of the oral cavity (*Pedersen et al., 2002*). This is most clearly appreciated in disease states which result in hypofunction where patients experience difficulty chewing and swallowing food, speaking and are susceptible to oral infections that result in dental caries (*Melvin, 1991*). Fluid secretion from the salivary gland requires the trans epithelial movement of Cl$^-$ across polarized acinar cells (*Foskett, 1990*; *Melvin, 1999*; *Melvin et al., 2005*; *Turner and Sugiya, 2002*). The polarization of exocrine cells is a consequence of the presence of tight junction protein complexes which functionally separate the transport machinery in the apical PM from those in the functionally equivalent, basal and lateral PM (collectively termed basolateral PM). This structural specialization allows distinct proteins to be localized in each membrane to promote vectoral ion and fluid movement (*Melvin et al., 2005*). The current, widely accepted model for fluid secretion posits that Cl$^-$ are transported against their electrochemical gradient by the Na$^+$/K$^+$/2Cl$^-$ cotransporter, NKCC1 at the basolateral plasma membrane (PM) (*Evans et al., 2000*) and subsequently enters the lumen of the acinus across the apical PM through TMEM16a chloride channels (*Romanenko et al., 2010a*). Cl$^-$ accumulation imparts a negative potential to the lumen relative to the basolateral PM, which draws Na$^+$ paracellularly through tight junctions (*Melvin et al., 2005*). Subsequently, water follows osmotically to generate the primary saliva which is modified by the ductal epithelia. An elevation in cytosolic [Ca$^{2+}$] following neurotransmitter release plays a central role in the stimulation of fluid secretion from salivary gland acinar cells (*Ambudkar, 2011*; *Ambudkar, 2016*; *Gallacher and Petersen, 1983*; *Putney, 1982*; *Weiss et al., 1982*). Acetylcholine (ACh) is the

primary neurotransmitter released following parasympathetic input to the major salivary glands after gustatory and olfactory stimulation (*Matsui et al., 2000*). ACh results in activation of muscarinic receptors, G-protein coupled stimulation of inositol 1,4,5 trisphosphate ($IP_3$) production and subsequent $Ca^{2+}$ release from endoplasmic reticulum stores via $IP_3$ receptors ($IP_3R$) (*Futatsugi et al., 2005*; *Lee et al., 1997*; *Pages et al., 2019*). The ER stores are subsequently refilled by store operated $Ca^{2+}$ entry across the PM, which is required for sustained salivary secretion (*Cheng et al., 2011a*; *Cheng et al., 2011b*; *Melvin et al., 1991*; *Takemura et al., 1989*). To stimulate saliva flow, the primary effector of the increase in $[Ca^{2+}]$ are TMEM16a channels localized in the apical PM (*Romanenko et al., 2010a*; *Arreola et al., 1996a*; *Arreola et al., 1996b*; *Begenisich and Melvin, 1998*). As $Cl^-$ moves across the apical PM, the membrane potential approaches the equilibrium potential for $Cl^-$; however, $Ca^{2+}$ also increases the activity of K channels, conventionally thought to be localized predominantly at the basolateral PM, to maintain the membrane potential to favor continued $Cl^-$ exit and secretion (*Maruyama et al., 1983*; *Nakamoto et al., 2008*; *Nakamoto et al., 2007*; *Romanenko et al., 2007*; *Romanenko et al., 2010b*).

Over the past several decades, the spatiotemporal properties of the agonist-stimulated $Ca^{2+}$ signal have been described in preparations of enzymatically isolated preparations of salivary gland acinar cells or lobules (*Park et al., 2012*; *Warner et al., 2008*) loaded with fluorescent $Ca^{2+}$ indicators and imaged by time resolved wide field or confocal microscopy. In total, these studies indicate that the agonist stimulated $Ca^{2+}$ signals exhibit characteristics which appear ideally tuned to activate these spatially separated $Ca^{2+}$-dependent ion channels required for secretion. For example, stimulation with muscarinic agonists, or more directly by photolysis of caged-$IP_3$, invariably results in the $Ca^{2+}$ signal initiating in the extreme apical ER followed by a rapid $Ca^{2+}$ wave which results in globalization of the signal throughout the cytoplasm (*Bruce et al., 2004*; *Bruce et al., 2002*; *Giovannucci et al., 2002*; *Larina and Thorn, 2005*; *Tojyo et al., 1997*; *Won et al., 2007*). The initiation sites, termed the 'trigger zone', correspond to the localization of the majority of type-2/3 $IP_3R$ (*Lee et al., 1997*; *Pages et al., 2019*), in ER in close apposition to TMEM16a in the PM. The intimate association of apical ER and apical PM is emphasized by the apparent co-localization of $IP_3R$ and TMEM16a by confocal microscopy, despite their localization in different membranes. Super-resolution STED imaging does resolve distinct distributions of the two proteins, such that the distance between apical ER and PM is estimated to be ~50–100 nm (*Sneyd et al., 2021*) and thus constitutes a region of privileged communication between messenger and effectors. The subsequent global spread is consistent with $Ca^{2+}$-induced $Ca^{2+}$ release through $IP_3R$ and ryanodine receptors (RyR) (*Won et al., 2007*) to allow the $Ca^{2+}$ signal to efficiently activate the majority of K channels present on the basolateral PM. At low agonist concentrations, the cycle of apical $Ca^{2+}$ release, global spread and subsequent $Ca^{2+}$ sequestration is repeated, such that temporally, the $Ca^{2+}$ signal is manifested as a series of oscillations with a relatively slow period (5–8/min) (*Bruce et al., 2002*). At higher concentrations of agonist, the initial globalization is maintained to produce a 'peak and plateau' type temporal response.

While in vitro studies of $Ca^{2+}$ signaling have provided a wealth of information to greatly inform our models of salivary secretion, there are general caveats which apply to these data. Primarily, the cells are studied out of a physiological context with no extracellular matrix, basement membrane and with probable disruption of cellular junctions as a result of the enzymatic isolation. A further complication is that isolated cells are typically exposed for a period of time to a constant concentration of agonist which in many cases is a non-hydrolysable analog, as opposed to the neural delivery of transmitter occurring in a pulsatile fashion that is subject in vivo to enzymatic degradation. Whether these factors influence the spatiotemporal characteristics of $Ca^{2+}$ signals in salivary glands is unknown. Additionally, since only indirect measurements of fluid secretion inferred from volume changes can be made in isolated acinar cell preparations (*Teos et al., 2016*), it is unclear how the concentration-dependent characteristics of stimulated $Ca^{2+}$ signals in vitro relate to the physiological stimulation of fluid secretion in vivo.

In this study, we generated mice expressing a genetically encoded $Ca^{2+}$ indicator specifically in exocrine acinar cells and imaged the $Ca^{2+}$ signals in submandibular glands in living mice by intravital multiphoton (MP) microscopy. Notably, stimulation of the endogenous neural input to the gland induces $Ca^{2+}$ signals with major spatiotemporal properties which are distinct from those observed in isolated acinar cells. Specifically, following moderate stimulation, apically localized $Ca^{2+}$ signals are generated which did not propagate globally. Furthermore, the apical signals consist of rapid

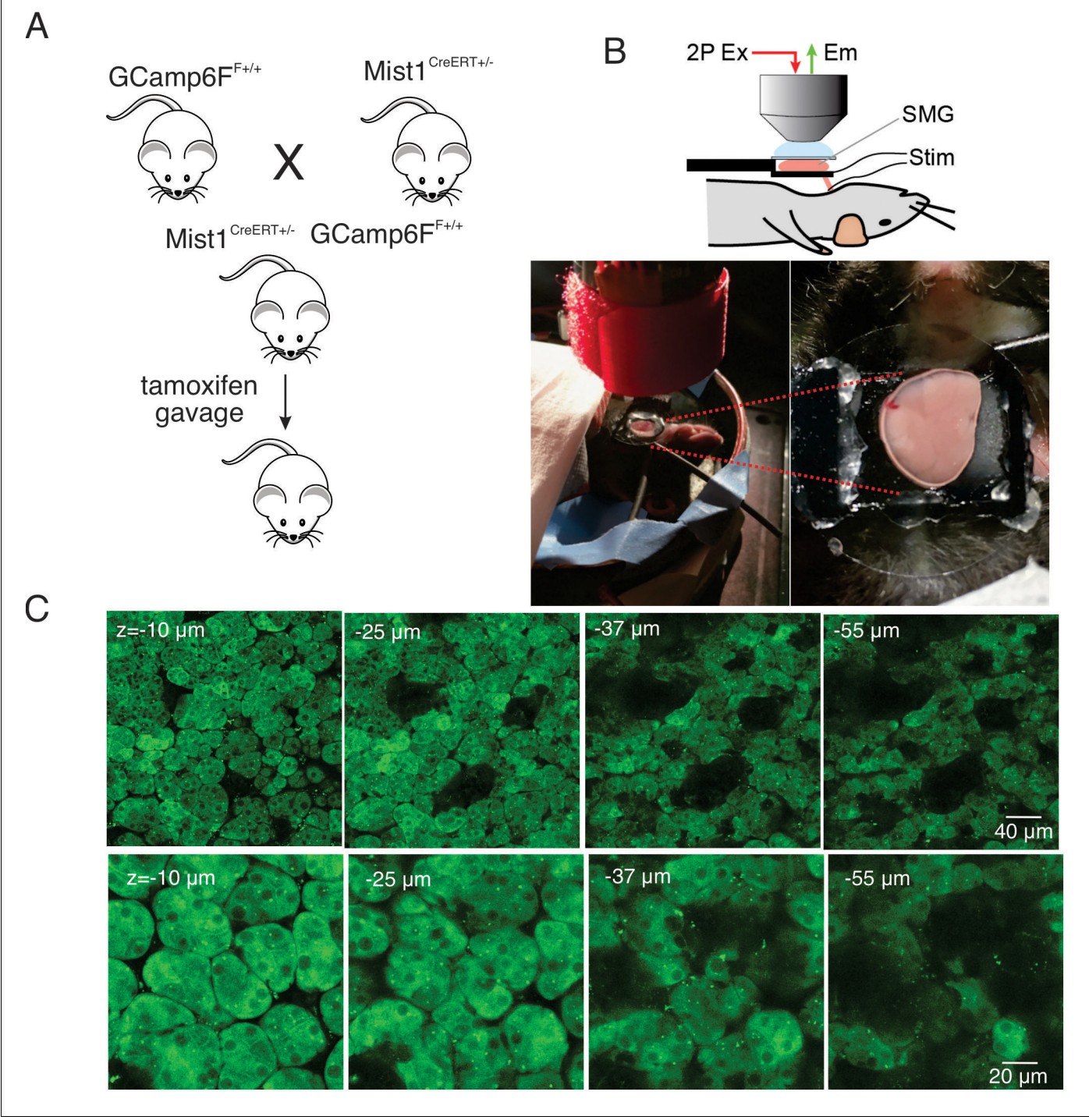

**Figure 1.** In vivo SMG Ca$^{2+}$ imaging. (**A**) Generation of transgenic mice expressing the GCaMP6f gene driven by Mist1 promotor. GCaMP6f is selectively expressed in acinar cells and appears uniformly through the cytoplasm but is largely excluded from the nucleus. (**B**) Animal preparation for in vivo imaging. A SMG was lifted and placed on a small platform, then held with a coverslip on top. Stimulation electrodes were inserted to a duct bundle that connect SMG to the body to stimulate nerves that innervate to SMG. (**C**) A series of z-projection images of SMG with GCaMP (green) expressed in acinar cells.

oscillations. At stimulation intensities optimum for secretion, a significant apical-basal standing Ca$^{2+}$ gradient was also established. These striking disparities suggest a modification of our previous model is necessary (*Pages et al., 2019*; *Sneyd et al., 2014*) to incorporate the characteristics of

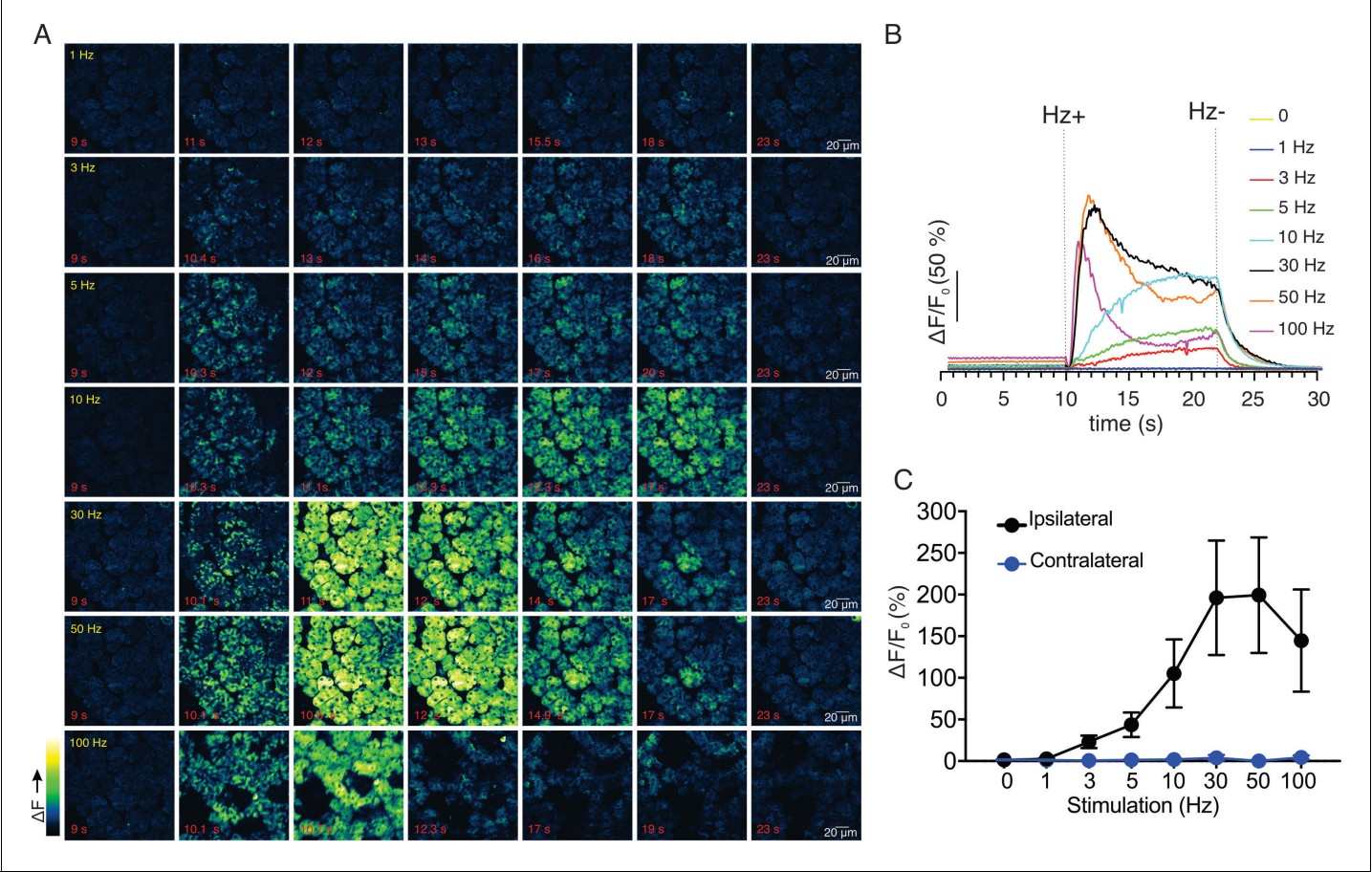

**Figure 2.** Ca$^{2+}$ signals evoked in response to nerve stimulation. (**A**) Time-course images of Ca$^{2+}$ (baseline fluorescence was subtracted) in response to 1–100 Hz stimulations. (**B**) Mean response of the entire field of view to stimulation at the indicated frequency. N = four fields from, four animals. (**C**) A summary of peak Ca$^{2+}$ increases during each stimulation. Stimulation was only to the ipsilateral SMG, as the contralateral SMG failed to respond. N = 4 fields from four animals. Mean ± sem. *p <0.05 *vs.* contralateral gland, t test.

The online version of this article includes the following source data for figure 2:

**Source data 1.** Source data associated with *Figure 2B and C*.

physiological Ca$^{2+}$ signals in vivo and thus provide a revised framework to better understand salivary fluid secretion.

## Results

To generate mice expressing a fluorescent Ca$^{2+}$ indicator in exocrine acinar cells, homozygous mice expressing the fast genetically encoded Ca$^{2+}$ indicator GCaMP6f (B6J.Cg-*Gt(ROSA)26Sor^{tm95.1(CAG-GCaMP6f)Hze}*/MwarJ) (*Chen et al., 2013*) floxed by a STOP cassette, were crossed with heterozygous tamoxifen-inducible Mist1 Cre mice (B6.129-*Bhlha15^{tm3(cre/ERT2)Skz}*/J) (*Maruyama et al., 2016*). The STOP codon was subsequently excised following tamoxifen gavage (*Figure 1A*). The expression of GCaMP6f was investigated in vivo in anaesthetized mice following surgery to expose the submandibular salivary glands (SMG). One SMG was lifted and placed on a small platform with a coverslip placed on top of the gland (*Figure 1B*). Bipolar stimulation electrodes were inserted to the duct bundle connecting to the imaged SMG tissue, in order to stimulate nerves innervating the SMG. A cover glass was placed on top of the SMG to gently hold the tissue and to maintain a saline solution bathing the gland. MP imaging in serial z sections through the gland revealed expression of GCaMP6f exclusively in acinar cells. GCaMP6f expression was uniformly observed in essentially all acinar cells while GCaMP6f expression was not evident in any other cell type such as blood vessels, nerves, myoepithelial cells or ductal cells. As expected by the size of GCaMP6f, the probe was

largely excluded from nuclei. Large voids with lack of expression of GCaMP6f were evident in z planes distal from the surface of the gland likely representing the localization of ductal structures (*Figure 1C*).

We next monitored changes in GCaMP6f fluorescence by intravital microscopy in live animals using MP excitation. Images were acquired at 30 frames per second (fps) as described in Materials and methods (each image a three frame average to result in acquisition at 10 fps). Under basal conditions, prior to stimulation, the cells were quiescent (*Figure 2B*). Electrical stimulation at various stimulus strengths delivered to nerves innervating the SMG (1–100 Hz, 12 s initiated at 10 s) induced increases in intracellular [Ca$^{2+}$] in acinar cells (*Figure 2A,B*). Following 1 Hz stimulation, a minority of the cells exhibited small, rapid changes in fluorescence, (n = 4 fields from four animals, *Figure 2A,B*), while most cells were refractory. Following higher frequency stimulation (3–10 Hz), the majority of the cells in the field responded and the amplitude of the [Ca$^{2+}$] signal increased as the stimulation strength was increased (*Figure 2A,B*). During the 12 s of stimulation, overall fluorescence gradually increased and immediately returned to baseline after the termination of the stimulation (*Figure 2A,B*). The maximum [Ca$^{2+}$] signal evoked during the 12 s stimulation progressively increased and was maintained from 1 to 10 Hz throughout the period of stimulation, but this trend did not hold following higher frequency stimulation (n = 4 fields from four animals, *Figure 2C*). Following stimulation at 30–100 Hz, the [Ca$^{2+}$] increase was not maintained for the duration of stimulation. Under these conditions, the [Ca$^{2+}$] rapidly peaked in the initial 1–3 s then decreased thereafter (*Figure 2A,B*), probably as a result of exceeding the maximal firing rate of the neurons innervating the SMG. Of note, the stimulation likely led to increased neural activity specifically, rather than stimulating acinar cells directly, as a consequence of current leak through the saline depolarizing nerve endings, as the contralateral SMG were unresponsive to even the highest 100 Hz nerve bundle stimulation of the ipsilateral gland (p > 0.1 compared to without stimulation, n=4 fields from four animals, *Figure 2C*).

To further confirm the physiological consequence of nerve stimulation, we monitored saliva secretion under identical stimulus conditions. There was no significant secretion activated by 1–3 Hz nerve stimulation for 1 min (p > 0.05 compared to the absence of nerve stimulation, n = four animals), but saliva secretion progressively increased following 5–10 Hz nerve stimulation (p < 0.05 compared to the absence of stimulation, n = four animals) (*Figure 3A*). Higher frequency stimulation did not result

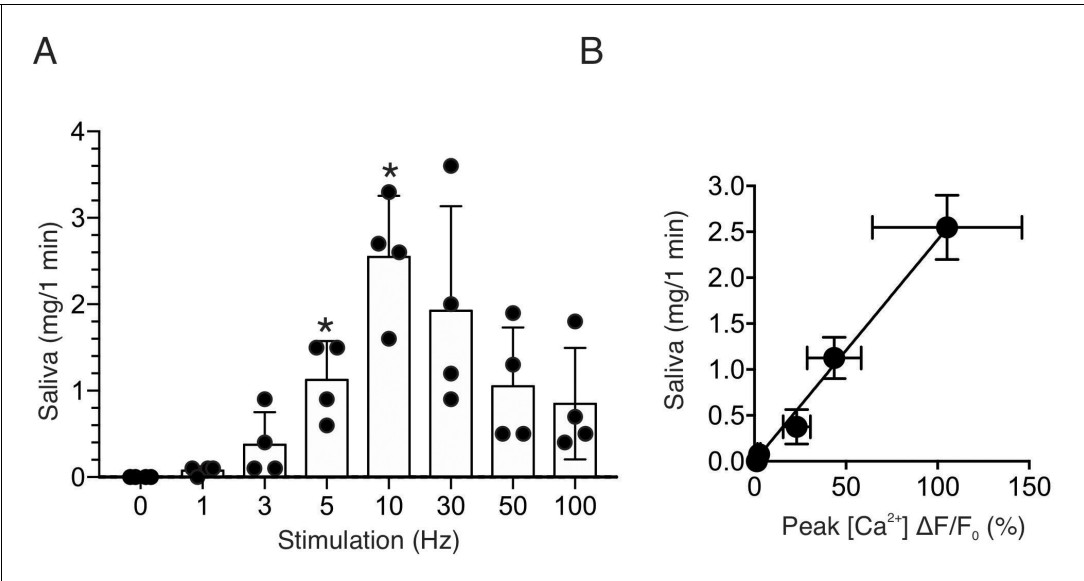

**Figure 3.** Saliva secretion following nerve stimulation. (A) A summary histogram of total saliva secretion following 1 min stimulations at the indicated frequency. N = four animals. Mean ± sem. (B) A correlation plot of peak Ca$^{2+}$ (shown in *Figure 2C*) vs. saliva secretion, which showed a linear regression of R$^2$ = 0.995 (black line) for stimulation frequency 1–10 Hz. *p <0.05 vs. no stimulation ANOVA with Dunnett test.
The online version of this article includes the following source data for figure 3:

**Source data 1.** Source data associated with *Figure 3A and B*.

in augmented secretion, likely reflecting failure to sustain the rise in [Ca$^{2+}$] (*Figure 2A*). At moderate stimulation intensity (0–10 Hz), there was a linear correlation between the peak [Ca$^{2+}$] increases by stimulation and saliva secretion (r$^2$ = 0.995, n = four animals, *Figure 3B*). These stimulus paradigms (1–10 Hz), therefore likely represent the range of physiological stimulation of both evoked Ca$^{2+}$ signals and the resulting saliva secretion and thus characterization of these signals, represent the focus of the remainder of the study.

Sustained saliva secretion requires both Ca$^{2+}$ release and subsequent Ca$^{2+}$ influx (*Ambudkar, 2011*; *Cheng et al., 2011b*; *Ambudkar et al., 2017*). While the former is dependent on IP$_3$-induced Ca$^{2+}$ release, the latter phase is ultimately dependent on Orai based channels (*Hong et al., 2011*). To provide evidence that similar mechanisms operate in vivo, experiments were performed stimulating glands at 10 Hz for a more extended period of time (~5 min) and evaluating the effects of an Orai channel blocker, GSK7975A (*Zhang et al., 2020*) on the evoked Ca$^{2+}$ signals. Stimulation resulted in an initial peak in [Ca$^{2+}$]$_i$ which decayed to a new plateau level which was maintained throughout the 5 min of stimulation (*Figure 4A*, n = four animals). Animals were subsequently injected with GSK7975A (20 mg/kg *ip*) and restimulated 20–70 mins later. No significant effect was observed on the magnitude of the initial peak, presumably because sufficient refilling of the ER Ca$^{2+}$ store had occurred during exposure to blocker. However, the plateau measured following 5 min of stimulation was significantly attenuated in drug injected animals *vs.* vehicle (*Figure 4A*, pooled data in *Figure 4B/C*) suggesting a requirement for Orai channel dependent influx for sustained Ca$^{2+}$ signals and secretion.

We next investigated the characteristics of Ca$^{2+}$ signals evoked by stimulation over this physiological range. Following nerve stimulation at low stimulation strengths (1–5 Hz), we observed both the recruitment of responding cells and an increase in the amplitude of the [Ca$^{2+}$] response (*Figure 5*). To determine more closely how the increased stimulation intensity impacts the characteristics of Ca$^{2+}$ responses to result in increased saliva secretion, a software-based work-flow was designed

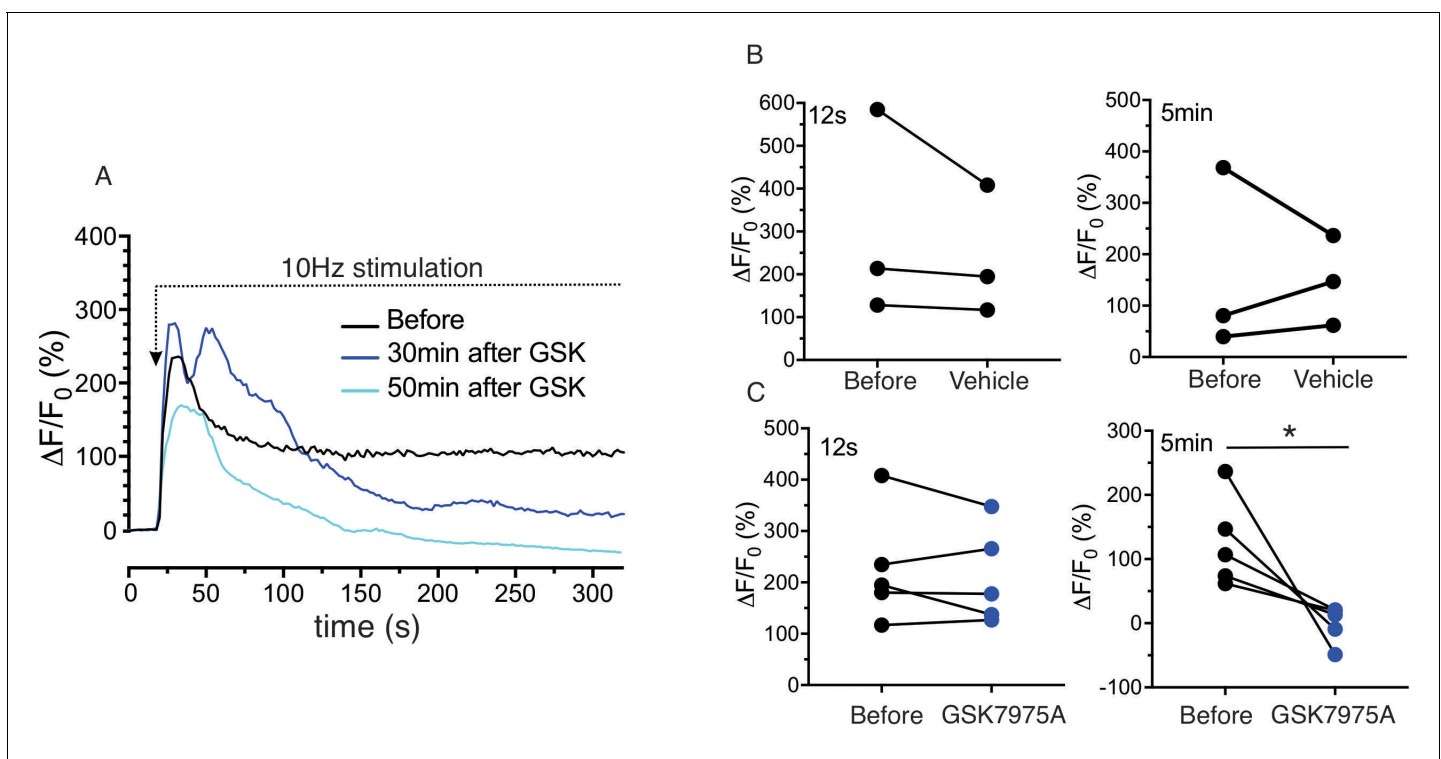

**Figure 4.** Sustained Ca$^{2+}$ signals are dependent on Ca$^{2+}$ influx through Orai channels. (A) Continuous 10 Hz stimulation results in an initial peak followed by a sustained plateau. In animals injected with the Orai channel blocker GSK7975A following the initial stimulation, subsequent stimulation results in a substantial reduction in the sustained phase of the response monitored following 5 min of stimulation. (B/C) shows the pooled data showing the magnitude of [Ca$^{2+}$] at initial peak (12 s after the initiation of stimulation, left) and sustained phase (5 min after the initiation, right) before and after 20–40 min of vehicle or GSK7975A administration. N = 3–5 from 3 to 5 animals. *p<0.05 paired t test.

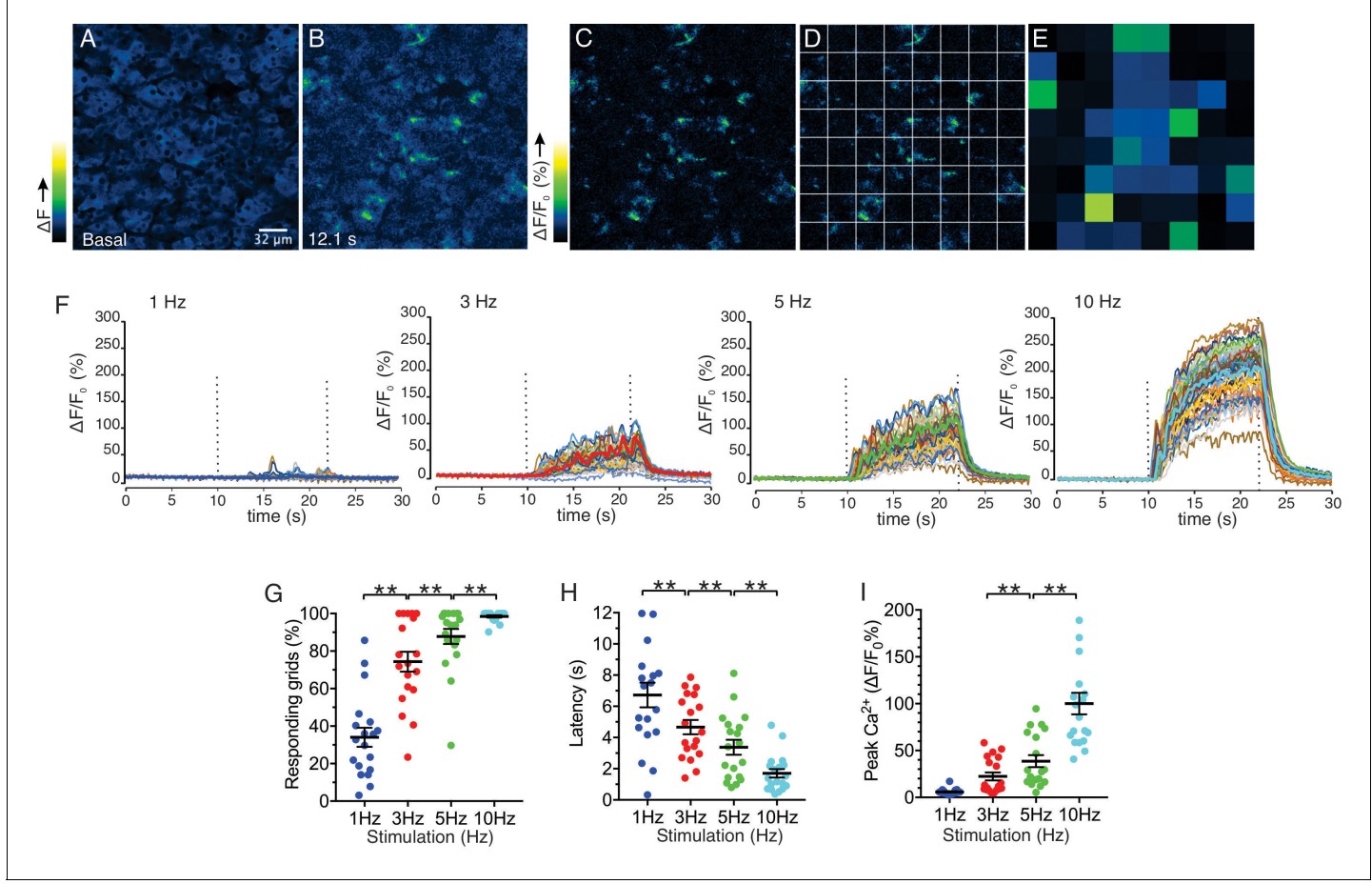

**Figure 5.** Spatial analysis of Ca$^{2+}$ responses to various stimulations. (**A–B**) An imaging field before (**A**) and during (**B**) a stimulation. (**C–D**) An imaging field was divided to grids of 32 x 32 µm, to yield 64 regions. (**E**) The average intensity per grid was obtained in each time frame. (**F**) Representative time-course plots of all 64 grids before, during, and after 1–10 Hz stimulations. Thick colored line represents the mean of the 64 grids for each stimulation. (**G**) A summary plot of average percent of responding grids in a field by the stimulations. N = 19 fields from eight animals. Mean ± sem. (**H**) A summary plot of average latencies to initiation of Ca$^{2+}$ increases in all grids. Non-responding grids were excluded for the data. Mean ± sem. N = 19 from eight animals. (**I**) A summary plot of average peak Ca$^{2+}$ increases in all grids. Mean ± sem. N = 19 from eight animals. **p<0.01 ANOVA with Tukey test.

The online version of this article includes the following source data and figure supplement(s) for figure 5:

**Source data 1.** Source data associated with *Figure 5G–I*.

**Figure supplement 1.** Grid subdivision of imaging field approximates the [Ca$^{2+}$] behaviors of lobules.

to post-process the image series. Imaged fields were sub-divided into 8 by 8 equal quadrants to yield 64 grid squares (*Figure 5C,D*). Each grid was 32 x 32 µm, and typically included a portion of an acinus consisting of 1–5 cells and was used to approximate the spatial and temporal behavior of individual cells or lobules in the field. In order to analyze the activity in each region, the image was then processed to show percent ΔF/F for each time-lapse frame in the sub-divided regions (*Figure 5D,E*). Using this general scheme, we compared information obtained from 8 x 8 grids, 16 x 16 grids and randomly manually selected regions of interest representing either acinar clusters or single acinar cells. Each scheme produced essentially equivalent data, ultimately validating the use of the 8 x 8 grid (*Figure 5—figure supplement 1*). At 1 Hz stimulation, the activity was sporadic, with the responding regions tending to give multiple responses while non-responding regions remained quiescent throughout the stimulus period (*Figure 5F*). Overall, 34.0 ± 5.1% (n = 19 fields from eight animals) of the sub-regions showed significant responses during the stimulation (*Figure 5G*). Moreover, the increase in [Ca$^{2+}$] in the responding regions exhibited a prolonged latency prior to the initial rise following stimulation (6.72 ± 0.79 s, n = 19, *Figure 5H*). Stimulation at higher frequencies

progressively recruited more grid regions and shortened the latency to the initial response (*Figure 5G,H*). At 10 Hz, virtually all regions responded to the stimulation (98.5 ± 0.6%, n = 19) with minimal latency (1.70 ± 0.27 s, n = 19) (*Figure 5G,H*). These data suggest that the increase in overall [$Ca^{2+}$] and secretion by rising stimulation strength was at least in part due to the recruitment of responding cells and a faster time for cells to respond. Further, the amplitude of the maximal increase in [$Ca^{2+}$] in the subdivided grids also increased as the strength of the stimulation was augmented (*Figure 5I*), indicating that enhancement of individual cellular [$Ca^{2+}$] responses also contribute to the overall increase of [$Ca^{2+}$] amplitudes.

We observed that increased stimulus strength resulted in both a higher number of responding regions and a higher peak [$Ca^{2+}$] amplitude in individual grids. We next examined whether grids which responded at lower stimulus intensity also strongly responded to higher frequency stimulation. Grids in an imaging field were ranked based on the peak [$Ca^{2+}$] evoked by 5 Hz stimulation and then re-sorted based on progressively increasing amplitude (*Figure 6A,B*). We subsequently sorted the data obtained from 1, 3, and 10 Hz stimulation from the identical field in the grid rank order established for 5 Hz stimulation (*Figure 6C*). This analysis demonstrated a clear pattern whereby, at each stimulus strength, particular regions of the field of view were most susceptible to stimulation, such

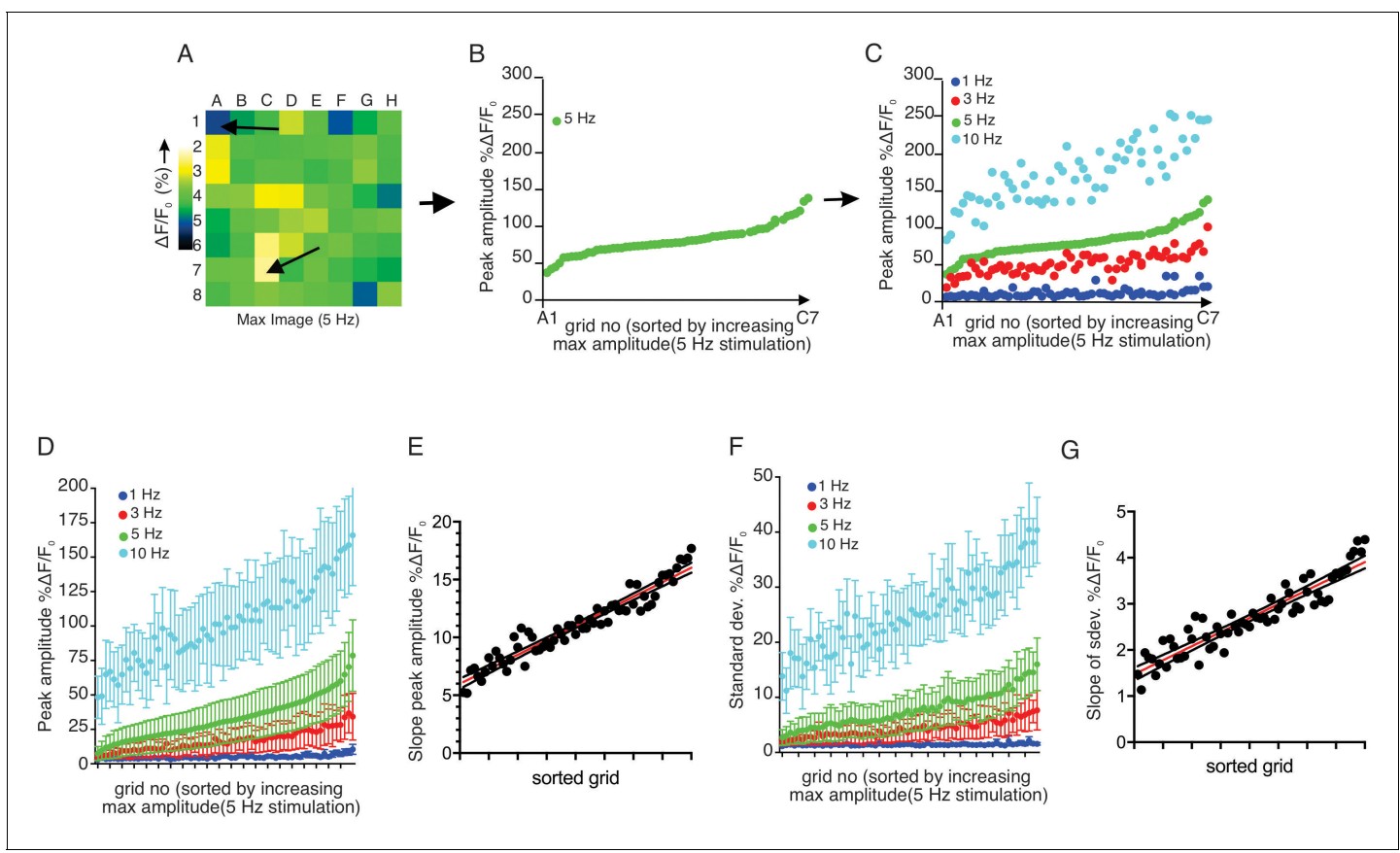

**Figure 6.** Each sub-divided field possesses unique sensitivity. (A) An example image of a processed imaging field showing peak intensities during a stimulation in each grid. The grids consist of lowest intensity (arrow, grid A1) to highest (arrow, grid C7). (B) A representative plot of peak $Ca^{2+}$ increases for each grid in a field to a 5 Hz stimulation. The grids in x axis were reordered from the lowest intensity to the left to the highest on the right. (C) Different stimulation strengths were applied to the same field, and the responses were plotted according to the order indexed by the responses to 5 Hz. (D) A summary plot of peak $Ca^{2+}$ increases in each grid sorted by responses to 5 Hz. Mean ± sem. n = 5 from four animals. (E) A correlation plot of sorted grid, from low peak $Ca^{2+}$ to high, vs. degree of $Ca^{2+}$ increases in response to higher stimulation. Linear regression with $R^2$ = 0.920. N = 5 from four animals. (F) A summary plot of standard deviations of $Ca^{2+}$ changes during stimulations in each grid sorted by responses to 5 Hz. N = 5 from four animals. Mean ± sem. (G) A correlation plot of sorted grids, from low peak $Ca^{2+}$ to high, vs. degree of standard deviation of $Ca^{2+}$ flux in response to higher stimulation. Linear regression with $R^2$ = 0.866. N = 5 from four animals.

The online version of this article includes the following source data for figure 6:

**Source data 1.** Source data associated with *Figure 6D–G*.

that grids giving the largest amplitude response at 1 Hz similarly responded by displaying the largest amplitude signals at 3, 5 and 10 Hz stimulation. (n = 5 fields from four animals) (*Figure 6D*).

The extent of the increase in [Ca$^{2+}$] following greater strengths of stimulation was not uniform throughout individual grids. High-responding grid regions showed linearly enhanced increases following stimulation (R$^2$ = 0.9197, n = 4 fields from four animals), while the increase in Ca$^{2+}$ signal following increased stimulation in less responsive regions was proportionately smaller (*Figure 6E*). This trend was not limited to the peak amplitude attained. During the 12 s stimulation, grid regions exhibited [Ca$^{2+}$] fluctuations superimposed upon an elevated plateau (*Figure 6F*). We evaluated the extent of this fluctuation by comparing the standard deviation of the ΔF/F signal occurring during the stimulation period. With this analysis, a higher standard deviation reflects greater variability in the stimulated change in [Ca$^{2+}$]. When the 5 Hz data was sorted according to the peak [Ca$^{2+}$] ranking as in *Figure 6D*, the variation of grids with 5 Hz stimulation also demonstrated that low-responding grids had smaller variability than high-responding grids (n = 4 fields from four animals) (*Figure 6F*). Likewise, the same trend was exhibited at each stimulation strength, with a linearly increasing rate of enhancement as intensity increased (R$^2$ = 0.8664, n = 4) (*Figure 6G*). Therefore, an increased fluctuation in the response is also a property of sensitive, highly responsive regions.

Stimulation of salivary fluid secretion is predominantly mediated through parasympathetic input as a result of the action of ACh released from nerve endings acting on muscarinic M3 receptors on acinar cells (*Matsui et al., 2000*). To gain direct evidence that neural release of ACh was responsible for the changes in [Ca$^{2+}$] observed, we systemically administered a cholinesterase inhibitor to block the clearance of ACh and effectively increase the local [ACh]. Physostigmine (0.1 mg/kg body weight *IP*) enhanced the peak amplitude of the Ca$^{2+}$ signals following stimulation (*Figure 7A–D*), indicating that acinar cell [Ca$^{2+}$] changes reflect the extracellular availability of acetylcholine released by stimulation. The augmentation in the Ca$^{2+}$ signal was not statistically significant at 1 Hz stimulation (p > 0.05, n = 5 fields from four animals), but became more prominent at 3 Hz with shorter latency, an increase in the number of responding grids and peak amplitude (p < 0.05 in % response and peak

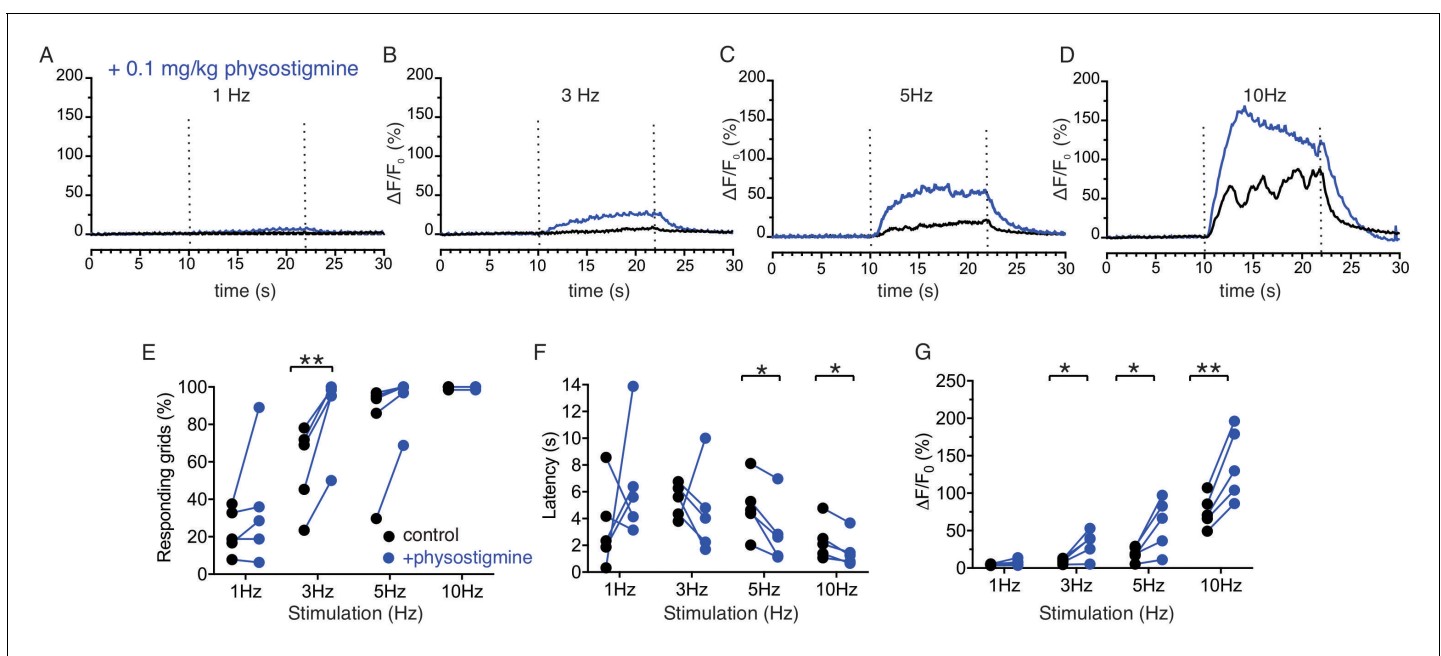

**Figure 7.** Enhancement of Ca$^{2+}$ responses by cholinesterase inhibition. (**A–D**) Representative average traces from 64 grids of changes in Ca$^{2+}$ responses in a same field before (black) and after (blue) physostigmine. (**E**) A summary plot of average percent of responding grids in a field by the stimulations in paired fields. N = 5 from four animals. Mean ± sem. (**F**) A summary plot of average latency to initiation of Ca$^{2+}$ increases in all grids in paired experiments. Non-responding grids were excluded for the data. N = 5 from four animals. Mean ± sem. (**G**) A summary plot in paired experiments of average peak Ca$^{2+}$ rise in all grids. N = 5 from four animals. Mean ± sem. *p<0.05; **p<0.01 Paired t test.

The online version of this article includes the following source data for figure 7:

**Source data 1.** Source data associated with *Figure 7E–G*.

ΔF/F, n = 5) and following 5 Hz stimulation (p < 0.05 in latency and peak ΔF/F, n = 5 fields from four animals) (*Figure 7E–G*). At 10 Hz stimulation, no further increase in the grid responsiveness or latency was observed, reflecting the fact that all grids were active and exhibited minimal latency at this stimulus strength, however the peak ΔF/F showed a robust enhancement (p < 0.01, n = 5) (*Figure 7E–G*). While these data do not rule out the involvement of additional neurotransmitters, they confirm a prominent role for ACh in mediating the increase in [Ca$^{2+}$] observed following neural stimulation.

While the previous grid-based analysis yielded important information relating to the general properties of the evoked signals within an imaged field, this approach was not suited to provide spatial information on a sub-acinus or lobule scale. To interrogate the subcellular spatial characteristics of the Ca$^{2+}$ signals evoked by neural stimulation, we generated standard deviation (SD) images from the stimulated portion of the image series. This analysis revealed that at all physiological stimulus strengths the increase in [Ca$^{2+}$]$_i$ was highly heterogeneous, with the largest increases in signal occurring in the apical aspects of acinar clusters, directly below the apical PM (*Figure 8—figure supplement 1*). *Figure 8A* illustrates a field in which an acinar cluster is highlighted prior to neural stimulation at 5 Hz (red box and image series in *Figure 8B*). In marked contrast to in vitro experiments, analysis of the image series demonstrated that the signal consisted of brief (< 2 s in duration), repetitive increases in [Ca$^{2+}$] (*Figure 8B*). The signals were highly localized, being initiated in, and almost entirely confined to the apical portions of individual cells, within a few μms of the apical PM (*Figure 8B–E*), with only very rare, small elevations in the basal aspects of the cell. This localization is illustrated by SD images of the whole field (*Figure 8D*) and a topographical SD representation of the highlighted cluster (*Figure 8D*) together with analysis of the fluorescent changes in time from regions of interest manually assigned to an apical Region of Interest (ROI) (red box), cytoplasmic ROI (blue box), and a basal ROI (green box) (*Figure 8E*). Consistent with tight coupling between acinar cells by gap junctions, signals often appeared to invade the apical region of neighboring cells (see images time stamped between 30.8 and 31.8 s in *Figure 8B*).

At 10 Hz stimulation, a rise in [Ca$^{2+}$] was also rapidly initiated in the extreme apical ER, close to the apical PM (imaging field in *Figure 9A* with red box showing the highlighted cluster depicted in *Figure 9B*). In contrast to lower stimulus strengths, Ca$^{2+}$ signals were observed in the basal aspects of the cell (*Figure 9B–E*), but of considerably lower amplitude, approximately 20–30% of that observed in the apical domain. Effectively a standing apical-basal [Ca$^{2+}$] gradient was established, as shown in the SD image of the field and the topographical representation of the SD of the highlighted region (*Figure 9C/D*), together with the analysis of manually positioned representative apical (red ROI), cytoplasmic (blue ROI) and basal regions (green ROI) (*Figure 9E*) to illustrate the magnitude of the gradient.

To further investigate this spatial heterogenicity, we performed MP line scans following stimulation at 3–10 Hz stimulation where the scanned line was orientated to interrogate changes in fluorescence originating in the apical region extending through the nucleus to the base of an individual cell. *Figure 10A* shows a representative field with the position of the scanned line. *Figure 10BI,CI and DI* show the changes in line fluorescence over time evoked by 3, 5, 10 Hz stimulation, respectively. Consistent with the previous data, following 3 and 5 Hz stimulation the changes in fluorescence were largely confined to regions close to the apical PM of the cell (defined as a 3 μm line in the kinetic from apical, cytoplasmic, and basal ROIs shown in *Figure 10BII and CII* and pooled data in *Figure 10E/F*). Stimulation at 10 Hz again resulted in large amplitude apical signals which diminished rapidly, such that the magnitude of the signal was ~25% of the apical signal in the extreme basal aspects of the cell distal to the nucleus (*Figure 10DI*, kinetic in 10 D$_{II}$ and pooled data from 11 line scans from 3 animals in 10E/F). The apical-basal gradient was further analyzed by plotting the amplitude of the maximum fluorescent change across the line profile which further reinforces that at each stimulus intensity the signal rapidly decays away from the apical initiation sites to the nucleus and is ~25% of the apical signal in the extreme base of the cell (*Figure 10*. B$_{III}$, C$_{III}$ and D$_{III}$). In total, these data demonstrate that in contrast to experiments performed in in vitro preparations, physiological stimulation results in prominent, apical Ca$^{2+}$ signals which do not propagate and globalize to result in a homogeneous [Ca$^{2+}$]$_i$ throughout the cell, such that steep apical-basal gradients are established.

To further analyze the neurally evoked apical Ca$^{2+}$ signals, Python software scripts were written in the Jupyter lab notebook environment to automate the designation of apical ROIs from masks

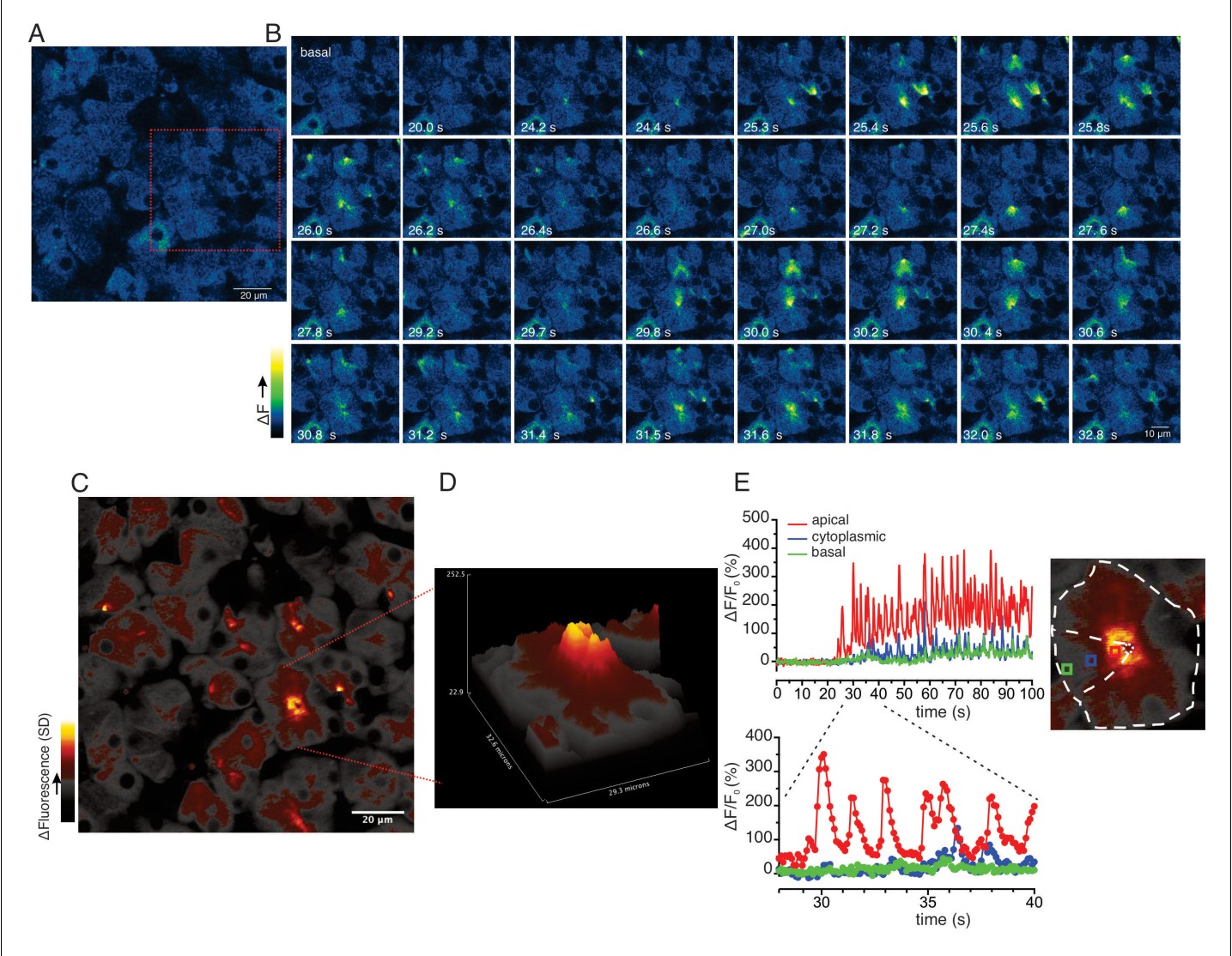

**Figure 8.** Ca$^{2+}$ signals are restricted to the apical region during moderate stimulation. (**A**) A representative imaging field prior to stimulation. (**B**) Time-course images of a field containing an acinar cluster (red box in panel A) showing active Ca$^{2+}$ signaling in the extreme apical portion in the cells. (**C**) A processed image mapping standard deviation values during stimulation period in the field shown in panel (**A**). (**D**) A 3D plot of the SD image of the acinar cluster in **B**, showing large fluctuations in [Ca$^{2+}$] only in the apical domain with minimal Ca$^{2+}$ signal transmission to the basal aspects of the cell. (**E**) Plot of Ca$^{2+}$ changes in a single cell from the apical domain (red box in the in the right image), cytoplasmic (blue box in the right image) and basal (green box in right image).

The online version of this article includes the following figure supplement(s) for figure 8:

**Figure supplement 1.** Standard deviation images generated from the image series for the period of stimulation in a single represented imaged field.

generated from average or SD images. These apical ROIs were then subsequently used to quantitate the magnitude and frequency of signals within these regions (see Materials and methods for details). *Figure 11A* shows an imaging field depicting the stimulated-basal average image for the field generated during the 12 s of stimulation at 5 Hz from which 79 apical ROIs were generated from masks shown in *Figure 11B/C*. *Figure 11D* shows representative responses from 4 of the 79 ROIs to stimulation over the range of physiological stimulus strengths. Consistent with the previous grid analysis, few apical regions responded to 1 Hz stimulation while the majority of ROIs responded to stimulation at 5–10 Hz (see *Videos 1–4*). At both 3 and 5 Hz stimulation, individual apical ROIs typically responded by generating a rapid train of oscillations. A peak counting algorithm was used to analyze the frequency of the oscillations for each ROI. This demonstrated that the frequency of oscillations

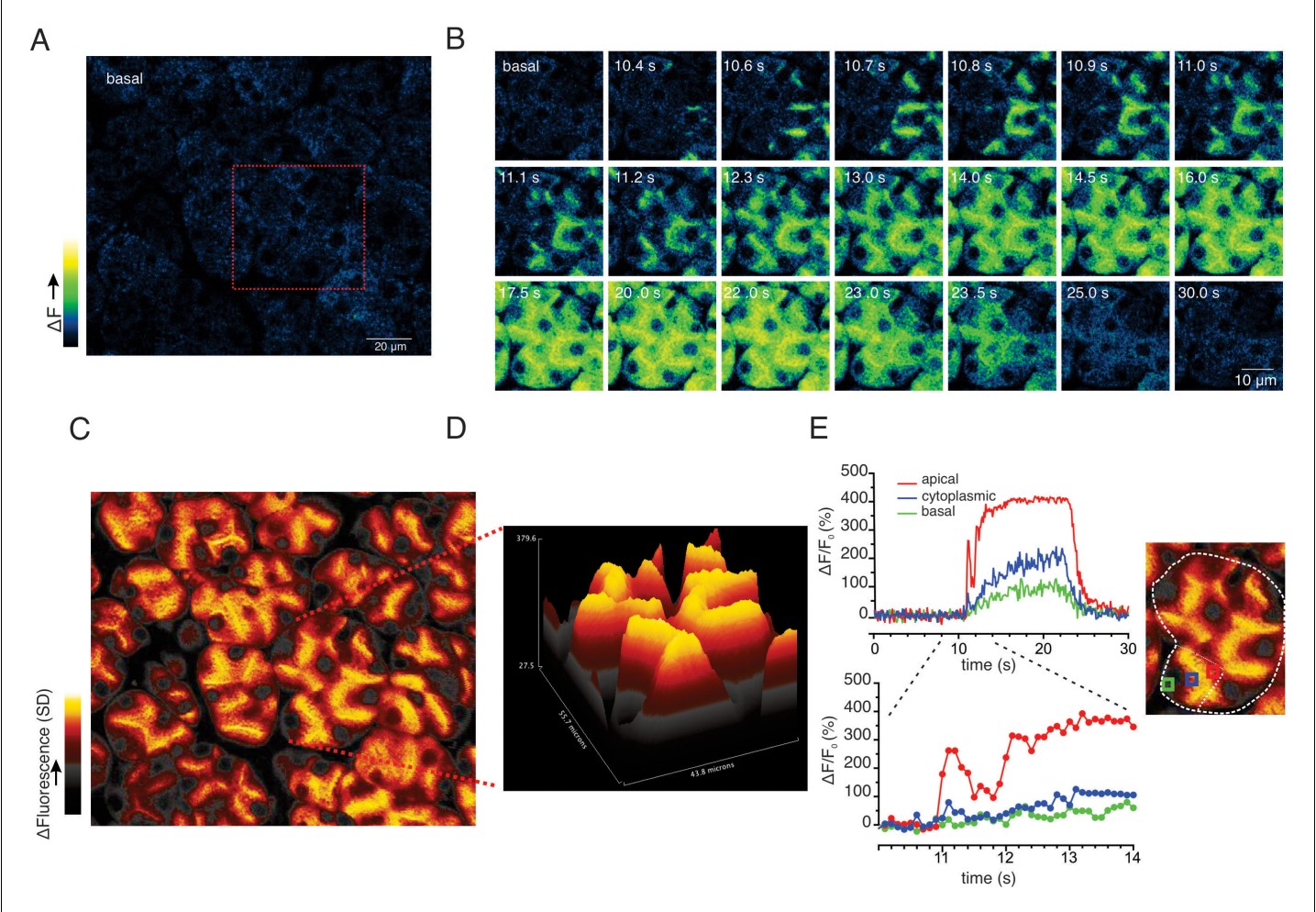

**Figure 9.** Apical to basal [Ca$^{2+}$] gradients are established during optimal stimulation. (**A**) A representative imaging field prior to stimulation. (**B**) Time-course images of a field containing an acinar cluster (red box in panel A) depicting predominant Ca$^{2+}$ changes in the apical portion of cells with smaller changes in the basal regions. (**C**) A processed image mapping standard deviation values during the stimulation period in the whole field shown in panel (**A**). (**D**) A 3D plot of the SD image for the acinar cluster shown in (**B**). (**E**) Tracings of Ca$^{2+}$ changes in the apical domain (red box in the image at right), cytoplasmic (blue box in the image on the right) and basal aspects of a single cell from this cluster (green box in the image at right).

was markedly faster (~1 Hz) when measured in vivo than previously described in isolated cells (~0.2 Hz, this study, *Figure 12*) and further modestly increased between 3 and 5 Hz stimulation (*Figure 11E*; 79 ROIs experiment shown in *Figure 11A–C* and pooled data from 13 individual fields from five animals (*Figure 11F*)). In contrast to the robust correlation between saliva secretion and the magnitude of the evoked Ca$^{2+}$ signal, the relationship between secretion and the frequency of Ca$^{2+}$ oscillation was less strong (R$^2$ = 0.824, *Figure 11G*).

Although GCaMP6f and fura-2 have similar affinities (*Badura et al., 2014*; *Lewis and Cahalan, 1989*), a possibility exists that the marked differences in the spatiotemporal properties of Ca$^{2+}$ signals observed in vivo vs. reported in vitro are a function of the genetically encoded indicator or alternatively because the vitro studies were largely performed at room temperature. We therefore prepared small acinar clumps from Mist1$^{+/-}$ x GCaMP6f$^{+/-}$ animals by enzymatic digestion and monitored Ca$^{2+}$ signals stimulated by the muscarinic agonist carbachol (CCh) by wide field imaging at 37° C. Individual acinar cells failed to respond to 5 nM CCh; however, at 25 nM CCh single cells within an acinus responded by evoking repetitive Ca$^{2+}$ transients (*Figure 12*) with a frequency ~0.2 Hz. The period of the transients tended to increase with increasing CCh concentrations (*Figure 12B*). In response to increasing concentrations of CCh, the initial peak increased (*Figure 12C*, n=14–32 cells, three animals) and the oscillations dampened to generate an elevated plateau. Oscillations evoked

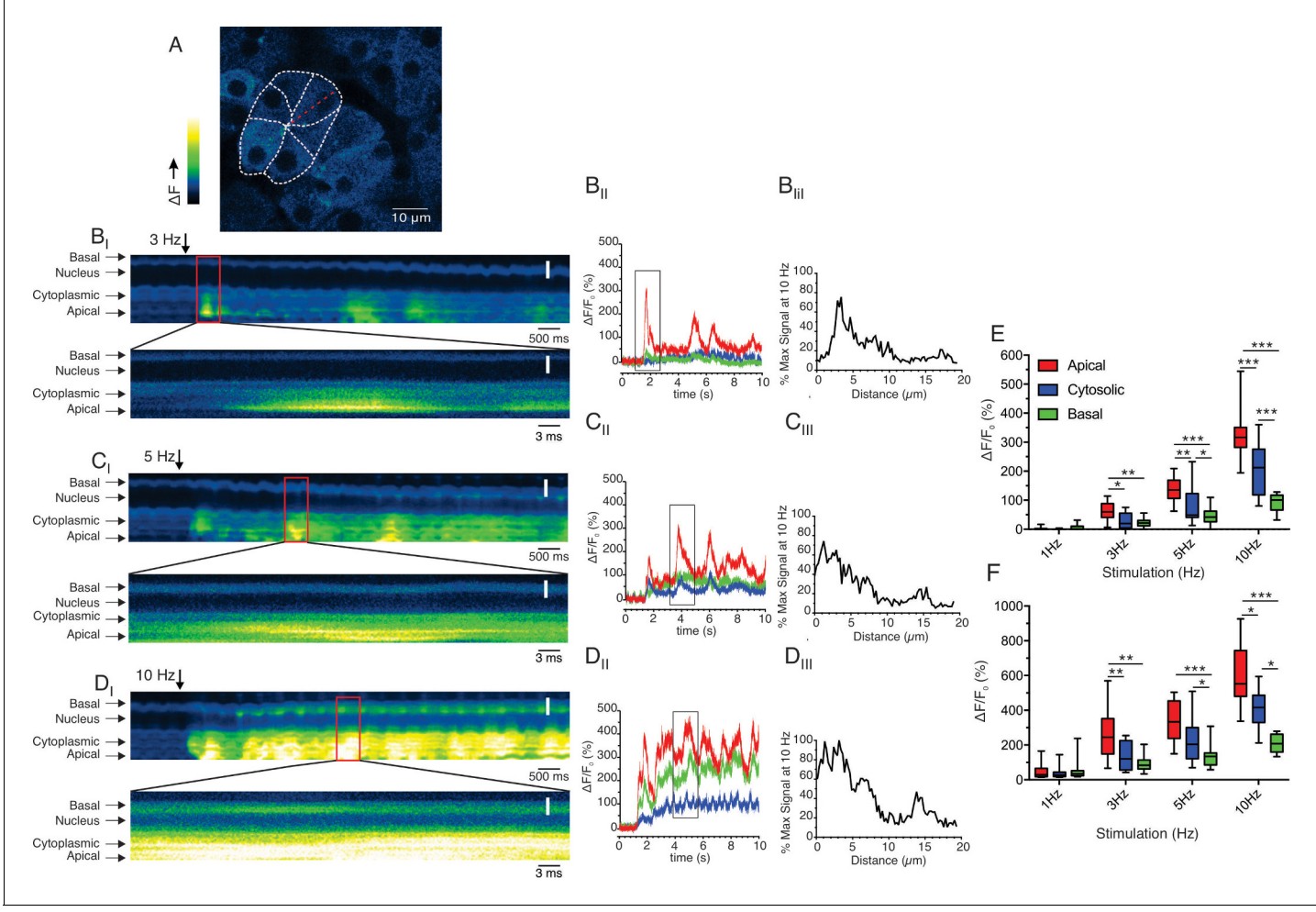

**Figure 10.** Apical-basal line scans reveal Ca²⁺ signal spatial homogeneity. (**A**) Shows a field of view where the boundaries of an acinus and individual cells have been delimited. The red line indicates the progression of the scan line. (**B_I, C_I and D_I**) Show consecutive lines stacked in time from left to right for 3, 5 and 10 Hz, respectively. The lower image for each stimulation is an expanded region encompassing the maximum increase in fluorescence for each stimulation. (**B_II, C_II and D_II**) Show kinetic plots for a 1 pixel 3 μm line placed in the apical, cytoplasmic (prior to the nucleus) and basal (distal to the nucleus) for 3, 5, 10 Hz stimulation, respectively. (**B_III, C_III and D_III**) Show the profile of the fluorescence along the scan line for the maximum increase in fluorescence for each stimulation frequency expressed as a % of the maximum fluorescence observed at 10 Hz. (**E**) Shows the pooled data depicting the average increase in apical, cytoplasmic, and basal ROIs at each stimulation frequency. (**F**) Shows the pooled data for maximal Ca²⁺ in apical, cytoplasmic, and basal ROIs at each stimulation frequency. White scale bar is 5 μm. ***p<0.001; **p<0.01,* <p< 0.05. Two-way ANOVA with multiple comparisons.

The online version of this article includes the following source data for figure 10:

**Source data 1.** Source data associated with *Figure 10E–F*.

even by threshold concentrations of CCh were initiated below the apical PM and invariably globalized with all CCh concentrations (*Figure 12D–F* and *Video 5*). In experiments performed at room temperature in GCaMP6f expressing cells (~23°C), the spatiotemporal characteristics of the signal were identical to those previously reported with chemical indicators (*Figure 12—figure supplement 1A–C*). The peak [Ca²⁺]ᵢ evoked by CCh stimulation was similar at 37°C (*Figure 12—figure supplement 1D*), while the oscillation frequency was significantly faster than observed at room temperature (*Figure 12—figure supplement 1E*). In total, the spatiotemporal profile of the Ca²⁺ responses evoked in acutely isolated GCaMP6f expressing acini were strikingly similar to numerous reports documenting agonist-evoked Ca²⁺ responses with conventional chemical Ca²⁺ indicators and thus the properties of the genetically encoded indicator at physiological temperature per se do not explain the differences observed.

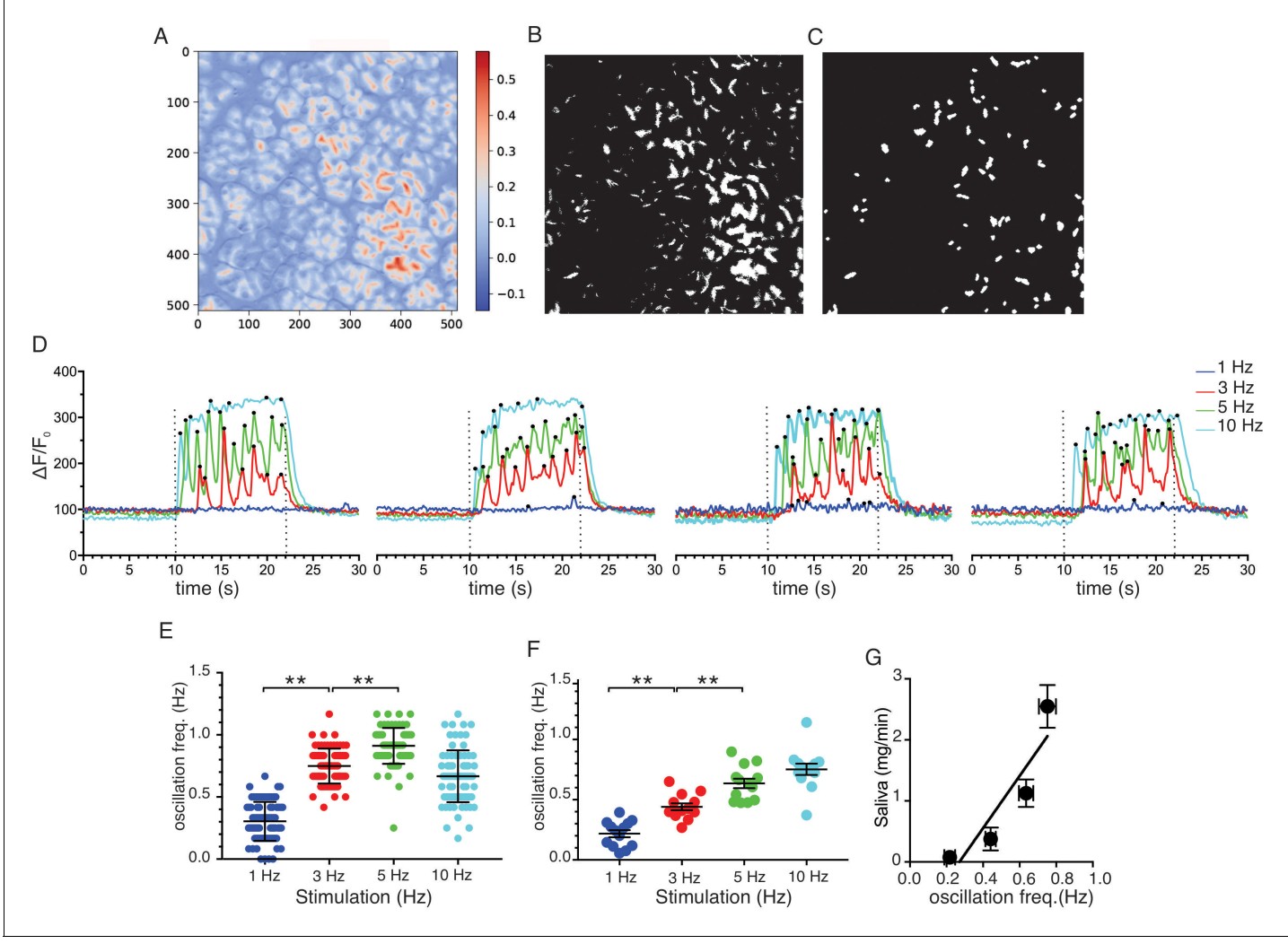

**Figure 11.** Ca$^{2+}$ Oscillation frequencies mildly correlates with stimulation strengths. (**A**) A representative image generated from averaging signal intensities during a 5 Hz stimulation, which highlights large fluorescence changes in the apical aspects of the cells. (**B–C**) Mask generated from the image in panel **A** using Python scripts running in the Jupyter lab environment, as described in Methods. (**D**) Representative Ca$^{2+}$ responses from 4 of the ROIs defined in the image in panel **C** following stimulation at 1–10 Hz. Black dots designate positions of oscillation peaks detected by automated peak detector software written in python. (**E**) A summary of oscillation frequency of all 79 ROIs from the image in panel A-D. Mean ± sem. (**F**) A summary plot of oscillation frequencies in response to 1–10 Hz stimulation. N = 13 fields from five animals. Mean ± sem. (**G**) A correlation plot of oscillation frequency vs. saliva secretion (shown in *Figure 3A*), which showed a linear regression with R$^2$ = 0.824 (black line). ** p<0.01 ANOVA with Tukey test.

The online version of this article includes the following source data for figure 11:

**Source data 1.** Source data associated with *Figure 11E and F*.

We have previously constructed a series of mathematical models to help understand and explain our experimental results and those from other groups (*Pages et al., 2019*; *Sneyd et al., 2014*; *Almássy et al., 2018*; *Sneyd et al., 2017a*; *Sneyd et al., 2017b*; *Vera-Sigüenza et al., 2019*). Briefly, these multiscale models were based on cyclical global increases in [Ca$^{2+}$] activating spatially separated ion channels required for sustained secretion in a 3D collection of seven acinar cells forming an acinus. In these older models, apical initiation of the Ca$^{2+}$ signal resulted in activation of Cl$^-$ flux through TMEM16a Cl$^-$ channels and the Ca$^{2+}$ signal was subsequently transmitted across the cell cytoplasm by a process of Ca$^{2+}$-induced Ca$^{2+}$ release (via RyR or IP$_3$R). This led to periodic increases of [Ca$^{2+}$] in the basal regions of the cell and hence activation of basal Ca$^{2+}$ activated K$^+$ (KCa) channels. Activation of the KCa channels is critical for maintenance of the electrochemical potential driving the efflux of Cl$^-$ and thus secretion. However, the spatiotemporal information from

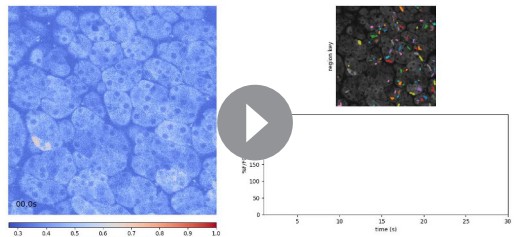

**Video 1.** Movie generated by Python scripts running in the Jupyter lab environment following stimulation at 1 Hz. Left panel shows $\Delta F/F_0$ image series. Bottom right panel depicts the change in $F/F_0$ for the apical ROIs automatically generated by the software described in *Figure 11*.

https://elifesciences.org/articles/66170#video1

the current in vivo data is obviously not compatible with these older models. We have therefore constructed a new model, as shown in *Figure 13*. The apical region is defined as the domain within 3 μm of the apical PM, which is itself demarcated by the tight junctions. This region now includes all the components necessary for saliva secretion, with KCa channels and Na/K ATPases in addition to the TMEM16a channels (*Figure 13*) which are poised to respond to $Ca^{2+}$ released from the apical ER, from $IP_3R$ that are situated in close proximity to the TMEM16A (*Pages et al., 2019*). We emphasise that, in the model, KCa channels and Na/K-ATPases are also present in the basal lateral membranes, and therefore the secretory machinery is not restricted to the apical PM. A more detailed description of the model can be found in methods and in *Pages et al., 2019*; *Vera-Sigüenza et al., 2019*; *Vera-Sigüenza et al., 2020*. The structure of the three-dimensional cell upon which our model is solved is shown in *Figure 14—figure supplement 1*, from where it can be seen that the apical membrane is a fingered structure that partially wraps around the cell, whereas the basal membrane (including the lateral membrane) extends right to the edge of the apical membrane.

*Figure 14A* shows the average apical $[Ca^{2+}]$ for a range of values of $V_{PLC}$ in the new model. $V_{PLC}$ controls the rate of production of $IP_3$, and is thus a proxy for frequency of neural stimulation. Simulations with a pulsatile $V_{PLC}$ (to mimic oscillatory neural stimulation) give identical qualitative results. As $V_{PLC}$ increases, the $[Ca^{2+}]$ oscillation frequency increases and the latency decreases. At high stimulation levels, the $[Ca^{2+}]$ oscillation frequency is approximately 0.5–1 Hz, and the oscillations are superimposed on an increasing baseline, which is due to a decreasing cell volume as a result of saliva secretion. These features all agree well with experimental data. One aspect of the data not reproduced by the model is the appearance of very low frequency baseline $[Ca^{2+}]$ spikes at the lowest stimulation frequencies. These baseline spikes are highly likely to be due to stochastic properties of the $IP_3R$, as described by Falcke and others (*Thurley et al., 2012*; *Thurley et al., 2014*; *Thurley et al., 2011*). Our model contains no stochastic components and is therefore, a priori, unable to reproduce such low-frequency baseline spiking.

*Figure 14*, panels B-D compare the new model with two older versions in which there were no apical KCa channels, but a $Ca^{2+}$ wave was propagated from the apical ER, throughout the cell, thus allowing for activation of basal KCa. *Figure 14B* shows saliva secretion (with VPLC = 0.008 μM/s) in the new model (red curve) compared with the

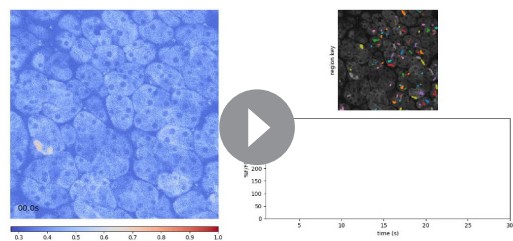

**Video 2.** Movie generated by Python scripts running in the Jupyter lab environment following stimulation at 3 Hz. Left panel shows $\Delta F/F_0$ image series. Bottom right panel depicts the change in $F/F_0$ for the apical ROIs automatically generated by the software described in *Figure 11*.

https://elifesciences.org/articles/66170#video2

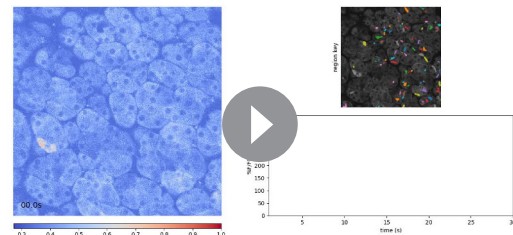

**Video 3.** Movie generated by Python scripts running in the Jupyter lab environment following stimulation at 5 Hz. Left panel shows $\Delta F/F_0$ image series. Bottom right panel depicts the change in $F/F_0$ for the apical ROIs automatically generated by the software described in *Figure 11*.

https://elifesciences.org/articles/66170#video3

old model of *Vera-Sigüenza et al., 2020* (green curve), which has no apical PM KCa but does exhibit a propagated [Ca$^{2+}$] signal from apical ER throughout the cell, and also compared to a version of the old model (blue curve) which has neither apical KCa nor a propagated [Ca$^{2+}$] wave. Note that the old model with no apical KCa is quantitatively different from the new model with no apical KCa. Fluid secretion in the new model with no apical KCa is slightly greater than that in the old model with no KCa, and is shown in panel F. These simulations show that absence of both a propagated [Ca$^{2+}$] wave and apical KCa channels significantly decreases saliva secretion. The associated [Ca$^{2+}$] traces, averaged from the apical and basal regions, as well as from the cytoplasmic region of the cell (representing a domain between these regions), are

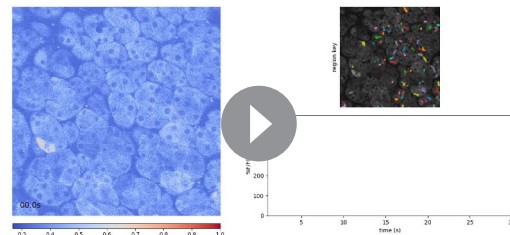

**Video 4.** Movie generated by Python scripts running in the Jupyter lab environment following stimulation at 10 Hz. Left panel shows ΔF/F$_0$ image series. Bottom right panel depicts the change in F/F$_0$ for the apical ROIs automatically generated by the software described in *Figure 11*.
https://elifesciences.org/articles/66170#video4

shown in *Figure 14*, C, D, using the same color scheme as in *Figure 14B*. In the new model (*Figure 14C*), basal [Ca$^{2+}$] is between 25% and 35% of apical [Ca$^{2+}$], as observed experimentally, while in the model of *Vera-Sigüenza et al., 2020*; *Figure 14D* basal [Ca$^{2+}$] is a significantly greater fraction of apical [Ca$^{2+}$]. The [Ca$^{2+}$] traces for the model with neither apical KCa nor a propagated wave are very similar to those shown in *Figure 14C*, and so are not shown explicitly. The intracellular spatial gradient of [Ca$^{2+}$] is illustrated by the model line scan shown in *Figure 14E*, which also agrees well with the experimental line scans, in that Ca$^{2+}$ is largely restricted to the apical region of the cell. The experimental data show more spatial inhomogeneity (due to the presence of the nucleus and the mitochondria), features which are not included in the model.

It is important to note that, in the model, of the basal region, includes the lateral PM and extends right to the edge of the apical region, which is a thin, fingered region. Thus, even in the absence of a propagated [Ca$^{2+}$] wave, diffusion of Ca$^{2+}$ from the apical region will still activate a fraction of the KCa channels. This is illustrated in *Figure 14F*, in the new model and for a range of stimulation levels, which shows how saliva secretion depends on the relative density of KCa channels in the apical PM. Even when there are no apical KCa channels (i.e. a relative density of 0), diffusion of Ca$^{2+}$ from the apical region can activate those basal KCa channels which are situated close to the apical membrane, and this is sufficient to cause significant fluid flow. Furthermore, consistently with previous theoretical work (*Palk et al., 2012*) there is an optimal density of apical KCa channels; if the apical membrane contains too high a fraction of the total number of KCa channels then fluid flow is hampered. Optimal fluid secretion occurs when the apical and basal KCa densities are approximately equal.

## Discussion

An increase in [Ca$^{2+}$] following neurotransmitter release from parasympathetic neurons is fundamentally important for the stimulation of salivary fluid secretion. It is also recognized that the spatiotemporal properties of intracellular Ca$^{2+}$ signals in exocrine cells are pivotal for appropriately activating ion channels necessary for the underlying process (*Almassy et al., 2012*; *Kasai and Augustine, 1990*; *Kidd and Thorn, 2000*). Despite a wealth of information documenting agonist-stimulated Ca$^{2+}$ signals, recorded in dissociated single cells, isolated acini or excised lobules, the properties of physiological Ca$^{2+}$ signals in vivo following endogenous neural stimulation have not been reported. We therefore generated transgenic mice that express a genetically encoded Ca$^{2+}$ indicator specifically in exocrine acinar cells and used MP microscopy in live mice to document the Ca$^{2+}$ signals generated following stimulation of the submandibular nerve which innervates the SMG. By establishing stimulation parameters optimum for fluid secretion, we have inferred the spatiotemporal properties of Ca$^{2+}$ signals that ultimately underlie the physiological stimulation of fluid secretion at the lobule level and subsequently at subcellular resolution.

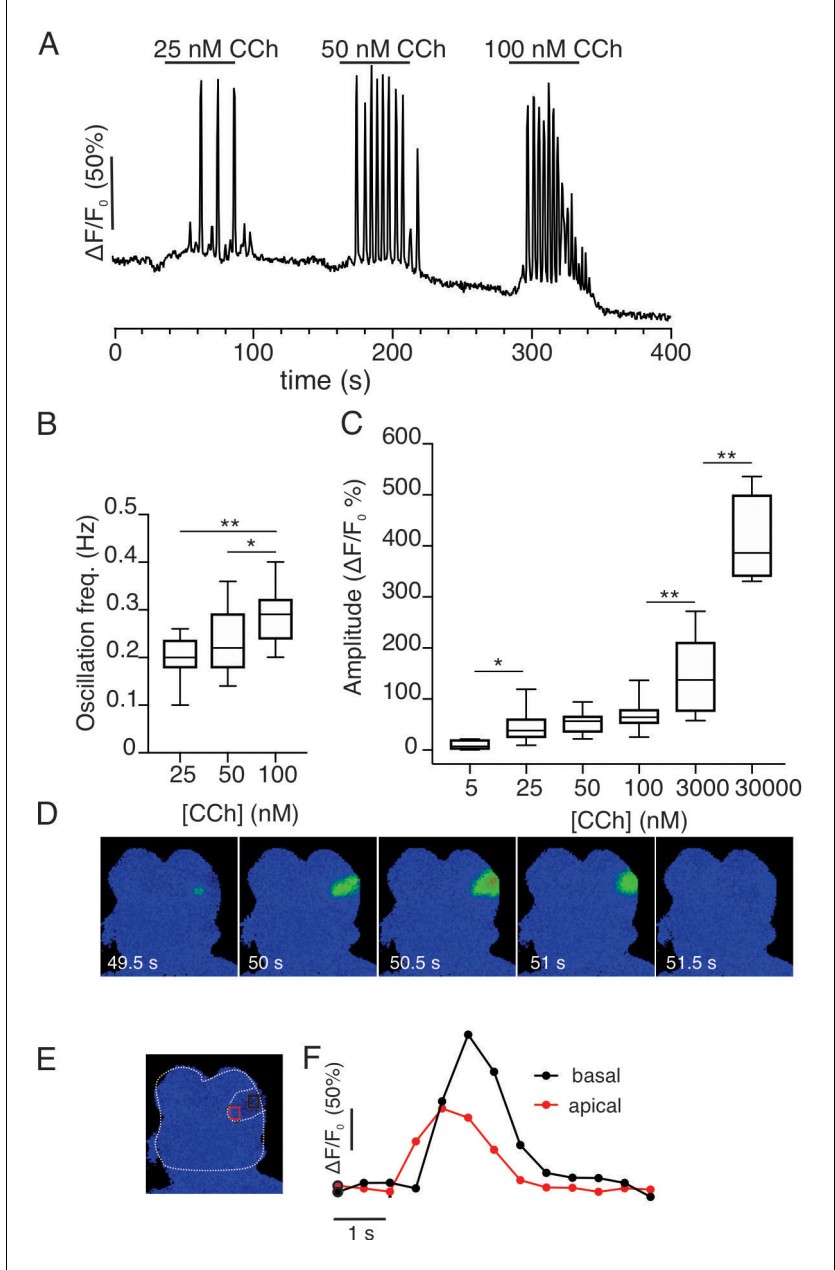

**Figure 12.** Imaging of acutely isolated acinar clusters from GCamp6f expressing SMG at 37˚C. (**A**) Cells were stimulated with increasing concentrations of CCh. (**A**) At concentrations greater than 25 nM, repetitive $Ca^{2+}$ transients were evoked in a minority of cells. (**B**) Pooled data depicting changes in oscillation frequency with increasing concentrations of carbachol. (**C**) Pooled data depicting changes in maximum amplitude of CCh-induced $Ca^{2+}$ signals. (**D**) Spatial changes in the $Ca^{2+}$ signal evoked by 25 nM CCh illustrating that the $Ca^{2+}$ signal is initiated in the extreme apical aspects of the cell and subsequently globalizes to reach the basal domain. (**E**) Shows a cartoon delimiting the cell shown in D and apical (red box) and basal (black box) ROIs. (**F**) Shows the kinetic illustrating the apical to basal $Ca^{2+}$ wave. * $p<0.05$; **$p<0.001$ One-way ANOVA.

The online version of this article includes the following figure supplement(s) for figure 12:

**Figure supplement 1.** In vitro imaging of acutely isolated acinar clusters isolated from GCamp6F expressing cells at room temperature.

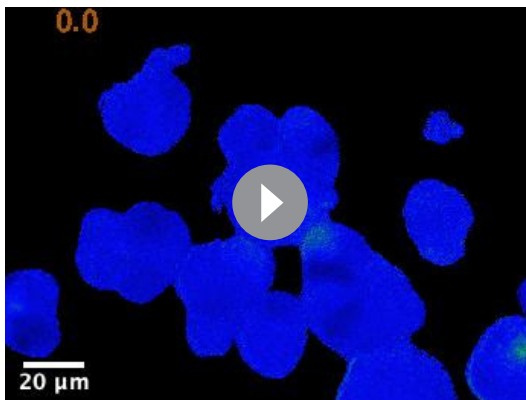

**Video 5.** Movie showing $\Delta F/F_0$ images from acutely isolated acinar clusters from GCamp6f expressing cells exposed to 25 nM (25–100 s), 50 nM (150–225 s), and 100 nM CCh (275–350 s).
https://elifesciences.org/articles/66170#video5

Following threshold stimulation, a small proportion of the imaged field responded after a prolonged latency by eliciting a single, or a few brief transients. These $Ca^{2+}$ spikes were initiated in the extreme apical ER, consistent with the localization of the majority of $IP_3R$ and previous in vitro data (*Lee et al., 1997*; *Pages et al., 2019*; *Won et al., 2007*). In stark contrast however, these signals did not globalize and were confined to a few μms distant from the apical PM. This threshold stimulation resulted in minor measurable fluid secretion. At greater stimulus strengths, where secretion was readily evident, an increasingly larger proportion of the field was recruited with shorter latency to yield multiple $Ca^{2+}$ transients with increasing peak magnitude. The increase reflecting an $[Ca^{2+}]_i$ rise in all cells within individual responding acini. Strikingly, these signals were again predominantly localized to regions immediately juxtaposed to the apical PM with limited propagation toward the basal aspects of the cell. Upon optimal stimulation for secretion, essentially the entire field was recruited to produce a sustained increase with periodic fluctuations superimposed on the plateau after a minimal latent period. Notably, while the affinity of the indicator may have masked a greater magnitude of apical $Ca^{2+}$ rise, or more prominent fluctuations in the signal at these higher stimulus strengths, a prominent apical to basal gradient was readily evident, such that the signal in the basal region reached only ~25–30% of that in the apical region. Further increases in the intensity of the stimulation resulted in a rapid peak followed by a gradual waning of the signal and consistent with the absence of a sustained $Ca^{2+}$ signal, the rate of fluid secretion was diminished. This reduced secretion likely results from a 'tetanic' stimulus where either neurons fail to fire during their refractory period, neurotransmitter release is exhausted or acinar cell $Ca^{2+}$ signaling is perturbed, during this likely non-physiological stimulation. Over the range of physiological stimulus strengths, the extent of fluid secretion was strongly correlated with the magnitude of the peak $Ca^{2+}$ rise, rather than necessarily the frequency of $Ca^{2+}$ transients. This conclusion is consistent with our previous modeling and experimental work that has indicated that the integrated increase in $[Ca^{2+}]$ during stimulation is the primary driving factor for efficient fluid secretion (*Sneyd et al., 2014*).

Local, non-propagating $Ca^{2+}$ signals are observed in isolated pancreatic acinar cell in response to $IP_3$, low concentrations of secretagogues and physiological concentrations of the gut hormone cholecystokinin (*Ashby et al., 2003*; *Thorn et al., 1993*; *Thorn et al., 1996*; *Thorn and Petersen, 1993*). It should be noted that the majority of these studies have been performed in whole-cell patch clamped isolated acinar cells. $Ca^{2+}$ buffering is established by the internal pipette solution in these experiments and thus could conceivably alter the spatial characteristics of the signal. Future in vivo experiments in the pancreas will address whether the spatial properties of $Ca^{2+}$ signals differ from those observed in acutely isolated cells. Nevertheless, the spatial characteristics of the $Ca^{2+}$ signal observed in vivo in the present study represents a major difference to previous work in isolated salivary acinar cells. The failure of the signal to propagate and standing $[Ca^{2+}]$ gradients observed are presumably established by SERCA, PMCA and mitochondrial buffering to efficiently reduce the local $[Ca^{2+}]$ to levels that do not support substantial CICR through $IP_3R$ and RYR that are presumed to be localized in the ER distant from the apical trigger zone. An obvious question is, what factors or mechanisms underlie these differences? At the level of the SMG lobule, the polarity of individual cells and their organization within an acinus is vital for function. The complex architecture is maintained by junctional complexes and interactions with the extracellular matrix and stromal cells (*Baker, 2010*). Clearly, these interactions may be disturbed when cells are isolated from an excised gland by enzymic digestion. An apparent consequence may be a disruption of the exquisite polarity of acinar cells. Perturbed localization of the $Ca^{2+}$ signaling machinery could disrupt the balance between $Ca^{2+}$ release and clearance to ultimately result in an alteration of the spatiotemporal

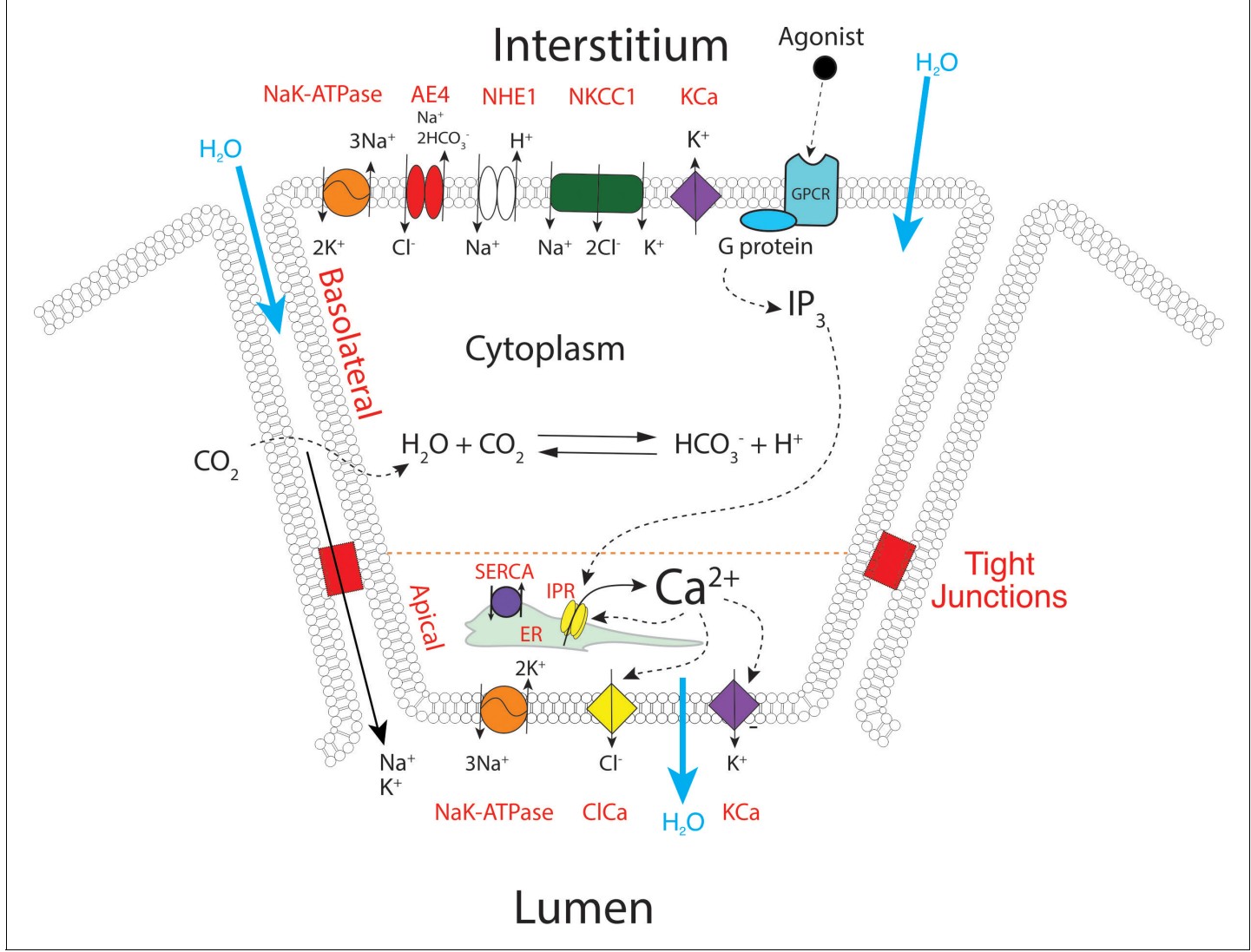

**Figure 13.** A proposed updated model for salivary secretion. The mathematical model is based on the fluxes and processes shown here. In contrast to previous models, the apical region now contains all the machinery needed for saliva secretion, included KCa channels and Na/K ATPases. $Ca^{2+}$ is released predominantly in the apical region, from $IP_3R$ that are situated in close proximity to TMEM16a, and there is no requirement for a propagated wave of increased $[Ca^{2+}]$ across the cell.

characteristics of the signal. Conceivably, alterations in the polarization of the ER, such that $IP_3R$ or SERCA are not maintained in the extreme apical ER (*Pages et al., 2019*), or a change in distribution of mitochondria (*Bruce et al., 2004*) to disrupt ER-mitochondrial junctions and effective $Ca^{2+}$ sequestration from release sites could contribute to the altered $Ca^{2+}$ signals observed in vitro.

A further important consideration when interpreting the characteristics of these $Ca^{2+}$ signals is the obvious difference in delivery of the secretagogue in these experiments. In contrast to superfusion of agonists in vitro, that results in a defined 'square' pulse of stimulating agent, in vivo, acinar cells likely experience a complex, stimulus intensity, and time-dependent gradient of neurotransmitter. In our studies, the concentration of ACh released from neurons would be predicted to rapidly increase and then decay as it is hydrolyzed by acetylcholinesterases. As the frequency of stimulation is increased, the residual concentration of ACh present in the vicinity of acinar cells would be predicted to increase with a 'saw-tooth' profile as the amount of ACh release exceeds the rate of degradation following more frequent, repetitive cycles of neurotransmitter exocytosis. In this scenario, when a threshold concentration of neurotransmitter is reached that generates sufficient $IP_3$, an increase $[Ca^{2+}]_i$, will be triggered. The time to reach this threshold would define the latent period for

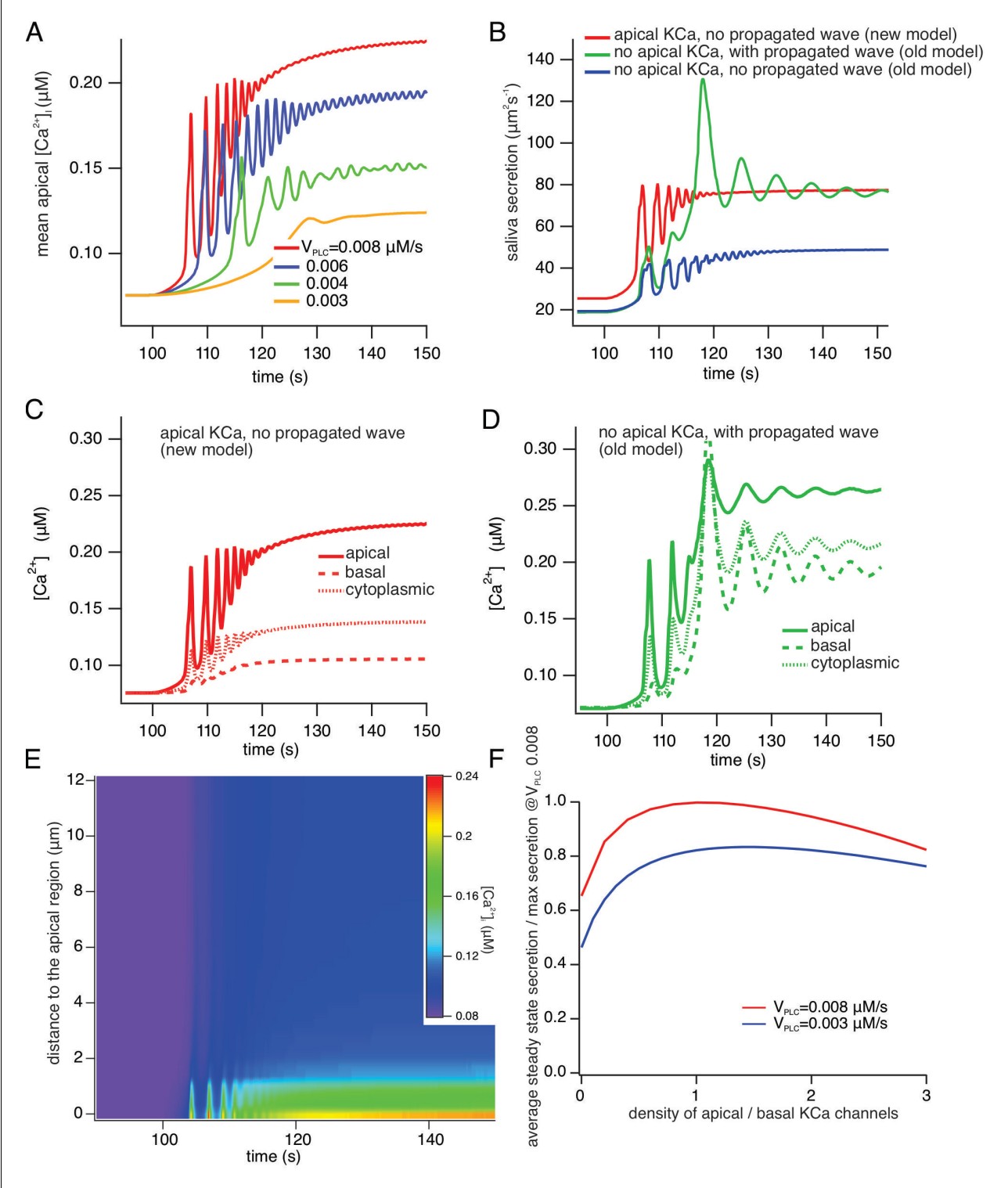

**Figure 14.** A comparison of model simulations incorporating new data. (**A**) Average apical $[Ca^{2+}]$ in the new model for four different values of $V_{PLC}$ (which is a proxy for stimulation frequency). The stimulus was applied at t=100 s. As the stimulus increases the model oscillations increase in frequency, and are superimposed on an increasing baseline, as seen experimentally. (**B**) Fluid secretion in the new model compared to two versions of the old model of *Vera-Sigüenza et al., 2019*. When the propagating intracellular $Ca^{2+}$ wave is removed from the old model (which has no apical KCa channels), secretion decreases significantly. (**C** and **D**) $[Ca^{2+}]$ traces averaged from three different regions of the cell in the new model and the old model of *Vera-Sigüenza et al., 2019*. (**E**) Line scan constructed from model simulations of the new model with $V_{PLC}$ = 0.008 μM/s. The stimulus was applied at t = 100 s. A line was drawn through the cell from the average of all the apical points to the average of all the basal points, and all grid points

*Figure 14 continued on next page*

*Figure 14 continued*

within 0.2 μm of this line were selected. The $Ca^{2+}$ traces from each of these grid points was then plotted as a color density plot, and laid side by side in order of distance from the apical region. The limited spatial resolution is a result of the limited spatial resolution of the grid, as only 36 grid points were close enough to the apical/basal line to be included in the line scan plot. The $Ca^{2+}$ responses are mostly confined to the apical region, although diffusion does cause a smaller basal response, as seen also in panel (F) Steady-state saliva secretion in the new model as a function of the relative density of apical KCa channels (relative to the density of basal KCa channels) at various vPLC. For all simulations the total number of KCa channels was the same. When there are no apical KCa channels (i.e. when the relative density is zero), fluid flow is decreased by approximately 35%. However, too many apical KCa channels also decreases secretion. Maximal secretion is attained approximately when the KCa channels have the same density in the apical and basal membranes. Note that, if the intracellular propagating $Ca^{2+}$ wave is removed from the old model it has a similar structure to the new model with no apical KCa channels. However, because of the presence of RyR in the old model, and because of the $Ca^{2+}$ dependence of PLC in the old model, there remain significant quantitative discrepancies between the old and new models even when they have a superficially similar structure.

The online version of this article includes the following figure supplement(s) for figure 14:

**Figure supplement 1.** The three-dimensional model cell upon which all the computations in this paper were performed.

that particular cell and the relative sensitivity to neurotransmitter will predicate whether an individual cell is responsive to a given stimulation. At optimum stimulation frequencies, the residual ACh concentration would further increase and remain above the threshold for the majority of acinar cells to respond. This would facilitate greater $Ca^{2+}$ release manifested as an increased peak and $Ca^{2+}$ oscillations by the entire field. Notably, neural stimulation does not simply 'pace' cells, as the frequency of oscillations does not correlate with the stimulus frequency. It is therefore likely, that the oscillations result from the inherent properties of $IP_3R$ and SERCA pumping activity but where the kinetics are influenced by the dynamic changes in neurotransmitter concentrations. This may contribute to the differences in oscillation frequency observed when comparing in vivo and in vitro data.

We observed considerable spatial heterogeneity in the responsiveness and peak amplitude of cells across the field of view. Specifically, there were sections of the field which exhibit the greatest sensitivity and largest responses at all stimulus intensities. Conceptually, this may reflect acini with expression of a higher density of cell surface receptors. Alternatively, highly responsive regions may represent cells physically closer to synapsing neurons and thus experiencing higher concentrations of ACh. The innervation of SMG lobules visualized in a reporter mouse expressing fluorescent protein in parasympathetic neurons has been reported to be fairly homogeneous (*Sheu et al., 2017*). Therefore, a further possibility is that spatial homogeneity arises in part because increasing numbers of axons are recruited to fire as stimulus intensity increases thus directly innervating a larger proportion of cells.

Clearly, our previous models describing how $Ca^{2+}$ signals promote salivary secretion are not entirely consistent with the new in vivo data presented here. Fundamentally, if there is no obligatory global, propagating intracellular $Ca^{2+}$ wave, activating the KCa channels in the basal PM would be predicted to be inefficient. Previously, however we have demonstrated, both functionally and physically, that the KCa channels, $K_{Ca}3.1$ ('IK') and $K_{Ca}1.1$ ('BK') together with Na/K ATPases are present in the apical PM of salivary acinar cells (*Almássy et al., 2018*; *Almassy et al., 2012*). Notably, we reported that $KCa_{3.1}$ colocalized with apical markers including $IP_3R3$ and Zona occludens-1. Further, focal, highly localized uncaging of $Ca^{2+}$, directly below the apical PM resulted in activation of a $K^+$ conductance with pharmacology consistent with $K_{Ca}3.1$. Additionally, apical uncaging of $Ca^{2+}$ evoked a smaller current in acini prepared from $K_{Ca}1.1$ knockout transgenic mice (*Almassy et al., 2012*). At the time these results were puzzling, as there was no obvious reason, according to the current model, why the apical PM should express these transport proteins. Nevertheless, a previous modeling study has suggested that expression of a proportion of KCa to the apical PM could optimize fluid secretion. The new data, together with the localization of apical KCa, combined with simulations by the new model, strongly suggests the physiological importance of the apical KCa and Na/K-ATPases. Specifically, as $[Ca^{2+}]$ elevations occur and are maintained at greater magnitude in close apposition to apical KCa channels, it would seem likely that their activation contributes to maintaining the electrochemical driving force for $Cl^-$ secretion. Given the localization of $IP_3R$ to apical ER, ~ 50–100 nm from the apical PM (*Sneyd et al., 2021*), a tight coupling between $Ca^{2+}$ release and activation of both major effectors would be reasonably predicted to promote efficient fluid secretion. The importance of apical KCa, while difficult to explicitly address experimentally, is supported by the new model which indicates that optimal secretion is achieved when KCa are expressed in both

the apical and basolateral membranes. On the other hand, since the primary saliva has a low $[K^+]$, this $K^+$ must be removed from the lumen. This is the proposed function of the Na/K-ATPase in the apical membrane.

Taken together, the present studies describe the spatial and temporal characteristics of physiological $Ca^{2+}$ signals driving salivary fluid secretion. Further, our findings highlight that caution should be exerted in extrapolating conclusions from ex vivo studies to physiological $Ca^{2+}$ signals and function in vivo. Finally, it is envisioned that the present studies will provide a framework for investigating if $Ca^{2+}$ signaling is disrupted in disease states such as Sjögren's syndrome or radiation-induced xerostomia which result in pronounced hyposecretion of saliva (*Teos et al., 2016*; *Baum et al., 2012*).

# Materials and methods

**Key resources table**

| Reagent type (species) or resource | Designation | Source or reference | Identifiers | Additional information |
|---|---|---|---|---|
| Strain, strain background (mouse) | B6J.Cg-Gt(ROSA)26 Sor<sup>tm95.1(CAG-GCaMP6f)Hze</sup>/MwarJ | Jackson Laboratory | RRID:IMSR_JAX:028865 | |
| Strain, strain background (mouse) | B6.129-Bhlha 15<sup>tm3(cre/ERT2)Skz</sup>/J | Jackson Laboratory | RRID:IMSR_JAX:029228 | |
| Chemical compound, drug | Physostigmine | Tocris Bioscience | 0622 | |
| Chemical compound, drug | GSK7975A | Sigma-Aldrich | 5.34351 | |
| Chemical compound, drug | Carbachol | Sigma-Aldrich | 1092009 | |
| Software, algorithm | FIJI/ImageJ | https://imagej.net/software/fiji/ | | |
| Software, algorithm | Prism | GraphPad | | |
| Software, algorithm | Jupyter Notebook | Jupyter lab https://jupyter.org/ | | |
| Software, algorithm | SciPy | https://www.scipy.org/ | | |
| Software, algorithm | scikit-image | scikit-image https://scikit-image.org | | |
| Other | Tamoxifen | Sigma-Aldrich | T5648 | |
| Other | Collagenase Type II | Worthington Biochemical | LS004204 | |

Tamoxifen, corn oil, ketamine, and Xylazine were purchased from Sigma Chemical, St Louis, Mo. Physostigmine was purchased from Tocris Minneapolis, Mn.

## Generation of mice expressing GCaMP6f in exocrine acinar cells

Adult female GCaMP6f<sup>flox</sup> mice (Jackson Laboratory; Jax 028865) were crossed with male heterogenous Mist1<sup>CreERT2</sup> (Jackson Laboratory; Jax 029228, kindly donated by Dr. Catherine Ovitt, University of Rochester) to generate Mist1$^{+/-}$ x GCaMP6f$^{+/-}$ transgenic mice. Tamoxifen (0.25 mg/g body weight) dissolved in corn oil was administered to mice at least 6 weeks old of either gender, by oral gavage once a day for 3 consecutive days. Imaging was carried out 1–4 weeks after the last tamoxifen administration. All animal procedures were approved by University Committee on Animal Resources (UCAR-2001-214E).

## In vivo MP imaging

The animal was anesthetized with Ketamine (80 mg/kg body weight, i.p.) and Xylazine (10 mg/kg body weight, i.p.). The animal was restrained with the ventral side up, and the animal's body temperature was maintained at 37°C by a heat pad during surgery and imaging. A submandibular gland was exposed by making an incision in the skin on the ventral side of the neck. Connective tissue around the gland was teased away using forceps to allow the gland to be raised away from the body but remaining attached by a duct/nerve/blood vessels bundle. A pair of tungsten wires (WPI, Inc) was inserted to the bundle as stimulation electrodes. The lifted gland was placed on a 10 x 13 mm custom build small holder situated directly above the neck so that the position of the gland was not influenced by movement as a result of breathing. The gland was held by a cover glass to flatten the surface and to keep tissue moist with Hanks salt solution between the holder and the cover glass (*Figure 1B*). A 25x water immersion lens (Olympus XLPlan N 1.05 W MP) equipped with an objective heater (OKOLab COL2532) kept at 37°C was utilized for MP imaging. An upright two-photon microscope system (FVMPE-RS, Olympus) with an excitation laser (InSight X3, Spectra-Physics) tuned at 950 nm and emission collected at 495–540 nm was utilized. Images were captured using Olympus FV31s-SW software at 30 fps framerate then averaged every three frames to achieve 10 fps image collection rate. Imaging depths between 10 and 55 µm from the surface of the gland were routinely utilized. PMT settings were fixed at 600V, 1x gain, and 3% black level, with excitation laser power adjusted per animal and according to imaging depth. Stimulation was generated by a stimulus isolator (Iso-flex, A.M.P.I.) set 5 mA, 200 µs, at the indicated frequency with train frequency and duration controlled by a train generator (DG2A, Warner Instruments). Images were captured typically for 30 s continuously. At least 1 min interval was given before imaging in the same field at a different stimulation strengths. In selected experiments, physostigmine (0.1 mg/kg body weight, i.p.) was administered to the animal on the microscope stage, at least 15 min prior to the imaging of post-physostigmine $Ca^{2+}$ signals.

For line scan imaging, an acinar cell cluster that arranged in the image plane such that that apical and basal sides of the acinar cell could be clearly identified and the imaging plane depth crossed near the center of the cell. A line was drawn from the apical end, passing through a nucleus, to the basal end of an acinar cell. The scanning speed was 1–1.6 ms per line for 10 s duration. The apical, cytosolic, and basal region were defined as 3 µm wide line within each image, and fluorescent intensity before stimulation for each region served as baseline intensity ($F_0$ = 0%).

## Saliva secretion measurement

An anesthetized animal with stimulation electrodes attached to the duct bundle were gently held with ventral side up. A piece of filter paper, roughly 2 x 10 mm, were weighed before placing it in the animal's mouth. A stimulation of 0–100 Hz was given to a gland for 1 min, and the paper was immediately removed and weighed. The difference of the weight of the paper represents the amount of saliva secreted out in the animal's mouth. A new paper was placed in the mouth for each stimulation train.

## Acinar cell isolation and imaging

SMG acinar cells were enzymatically isolated from 2- to 4-month-old, Mist1$^{+/-}$ x GCaMP6f$^{+/-}$ mice of both sexes. To isolate acinar cells, glands were extracted, connective tissue was removed, and glands were minced. Cells were placed in oxygenated dissociation media at 37°C for ~30 min with shaking. Dissociation media consisted of Hank's Balanced Salt Solution containing $CaCl_2$ and $MgCl_2$ (HBSS), bovine serum albumin (0.5%), and Collagenase Type II (0.2 mg/mL, Worthington). Cells were washed twice in HBSS with 0.5% BSA and resuspended in a HBSS solution containing 0.5% BSA and 0.02% trypsin inhibitor. Cells were then resuspended in imaging buffer (10 mM HEPES, 1.26 mM $Ca^{2+}$, 137 mM NaCl, 4.7 mM KCl, 5.5 mM glucose, 1 mM $Na_2HPO_4$, 0.56 mM $MgCl_2$, at pH 7.4) and seeded onto a coverslip to allow attachment of cells. Cells were then perfused with imaging buffer and stimulated with agonist. $Ca^{2+}$ imaging was performed using an inverted epifluorescence Nikon microscope with a 40 X oil immersion objective (NA=1.3) equipped with an environmental chamber allowing control of experimental temperature. Cells were excited at 488 nm from a monochromator, and emission was monitored at 530 nm. Images were captured every 500 ms with an exposure of 10 ms and 4 × four binning using a digital camera driven by TILL Photonics software. Image acquisition

was performed using TILLvisION software and data was exported to Microsoft excel and analyzed in Prism (Graph Pad).

## Image analysis

Amplitude, latency, and percent of responding areas in fluorescence fields were analyzed using ImageJ software (NIH). Standard Deviation (SD) Images were generated from the image time series by using the Stacks Z project function in image J. XY drift during the imaging, when present, was corrected by applying the Descriptor-based series registration plugin (Stephan Preibisch). The imaging field was divided into 8 by eight grids to yield 64 regions of interest (ROIs) of dimension 32x32 μm square. The average intensity of each ROI grid was generated for each frame. The average of first 100 frames prior to stimulation served as baseline fluorescence ($F_0$), and %$\Delta F/F_0$ ((F– $F_0$)/ $F_0$ x 100) was calculated using the Image calculator function, so that the converted 32-bit image series represents [$Ca^{2+}$] changes over time expressed as %$\Delta F/F_0$ in 8x8 alley of grids in the XY dimension (*Figure 5A*). The standard deviation of the baseline 100 frames from grids in each image series provided an estimate of the noise level. A change greater than four times the standard deviation from the $F_0$ value was considered as a stimulated $Ca^{2+}$ signal and the ROI therefore designating a responding grid. To calculate latency prior to a response the time at the first incident $Ca^{2+}$ signal meeting the criteria as a responding grid in each ROI after the initiation of the stimulation were recorded. Non-responding grids were excluded from this analysis. Statistical analyses were performed with paired t test, one-way ANOVA, and linear regression using Prism (GraphPad).

## Subcellular image analysis

A computer based, automated region-of-interest (ROI) detection method that specifically targets cellular regions with pronounced changes was developed since no off-the-shelf tool met our particular requirements to efficiently process the data (https://github.com/jrugis/cell_tools; *Rugis, 2021*; copy archived at swh:1:rev:c8fced9a9d8f3b7526a8f8eabfe547c618f2374b). The new software toolset consists of an assembly of Python (https://www.python.org) scripts deployed in a collection of Jupyter Lab (https://jupyter.org) 'notebooks'. The notebooks were designed to automate repetitive analysis steps and give flexibility to interactively explore and analyze the image data sets and makes use of image processing algorithms in the scikit-image (https://scikit-image.org) package. ROI were identified by generating a difference image whereby the average image prior to stimulation was subtracted from the average image during the period of stimulation on a pixel-by-pixel basis (*Figure 11A*). An initial mask was created by simple pixel intensity thresholding (*Figure 11B*). Subsequent filtering by binary dilation and binary erosion removes undesired small regions and smooths the remaining regions (*Figure 11C*). The notebook then generates plots of pixel intensity over time for each ROI generated and utilizes the same mask for images sets generated from the same field at differing frequencies of stimulation (*Figure 11D*). A further script block was generated to objectively automate the identification of peaks generated by stimulation. For peak processing several signal processing operations from the scipy (https://www.scipy.org) package were employed. The script employs a sequence of signal resampling, low-pass zero phase filtering, high-pass zero phase filtering and a generic peak detector followed by a mapping back into the original response data. Detected peaks for each stimulation frequency were plotted in the notebook as black dots overlaid on region summary plots as shown in *Figure 11D*.

## Model details

The model has two interconnected modules; a $Ca^{2+}$ oscillation module and a fluid secretion module. The two modules are coupled via changes in cell volume, and by the activation of KCa and ClCa by $Ca^{2+}$. The $Ca^{2+}$ oscillation module is a reaction-diffusion equation and is solved in a three-dimensional domain reconstructed from experimental structural data (*Vera-Sigüenza et al., 2020*). All ions except $Ca^{2+}$ are assumed to be homogeneously distributed in the cell, and are thus described by a system of ordinary differential equations. The apical membrane forms a fingered region that extends partially around the cell (*Figure 14—figure supplement 1*) and all the remainder of the membrane is assumed to be basolateral. No functional distinction is made between the basal and lateral membranes. TMEM channels are restricted to the apical membrane and are assumed to have a constant density there, and the other ion channels and/or transporters are assumed to be distributed on the

basolateral membrane with a constant density. KCa channels and Na/K ATPases are assumed to be distributed on both the apical and basolateral membranes. A first-order implicit-explicit finite element method is used to solve the reaction-diffusion for $[Ca^{2+}]$, and this is coupled to the ordinary differential equations for the other ions by a stepwise process in which the Matlab routine ode15s is first used to solve for the volume over a single time step, and that volume is then used to advance the reaction-diffusion equation for $[Ca^{2+}]$. The individual models for the various transporters, channels and exchangers are all given in *Vera-Sigüenza et al., 2020*. The model equations were solved in a single, uncoupled cell (cell four in the notation of *Vera-Sigüenza et al., 2020*. Similar results were obtained from all the other cells). The volume of the cell is taken into account, not by a full remeshing at each time step, but simply by a volume scaling factor in the reaction diffusion equation. The activity of the TMEM and KCa channels on a particular patch of membrane (described by a triangle in the finite element method) was assumed to be a function only of the mean $[Ca^{2+}]$ in the tetrahedron containing that surface triangle. Thus, the KCa are spatially distributed and activated in a manner that depends directly on the spatial distribution of $[Ca^{2+}]$ as a result of $Ca^{2+}$ release. We note that this means that $Ca^{2+}$ released from the apical region (which, in the model, means $Ca^{2+}$ release directly apposed to the surface triangles corresponding to the apical membrane) can diffuse a short distance to activate those KCa channels situated on the basolateral membrane close to the apical membrane. This is why the model exhibits substantial fluid flow even when there are no apical KCa channels; in the absence of apical KCa channels only a small proportion of the basolateral KCa channels will be activated by $Ca^{2+}$ diffusing from the apical region, but that is sufficient to allow for some fluid flow.

The $Ca^{2+}$ oscillation module is based on a closed-cell Class I model in which oscillations arise via sequential activation and inactivation of the $IP_3R$. $[Ca^{2+}]$ buffering is assumed to be fast and linear, and the $IP_3R$ model is taken from *Sneyd et al., 2017a*. $Ca^{2+}$ obeys a reaction-diffusion equation in the cell interior, but the effective diffusion coefficient of $Ca^{2+}$ is small (due to buffering) which allows for localized increases in $[Ca^{2+}]$ which are not propagated throughout the cell. Because the $IP_3R$ are situated 50–100 nm from the ClCa (which are on the apical membrane), release of $Ca^{2+}$ through the $IP_3R$ is modeled as a boundary flux term, which avoids the necessity for a high-resolution finite element mesh in the apical region, which would greatly slow the calculations. The diffusion coefficient of $IP_3$ is two orders of magnitude greater than that of $Ca^{2+}$, and thus $IP_3$ is effectively spatially homogeneous throughout the cell.

In the new model, all parameters and equations remain unchanged from those in the old model (*Vera-Sigüenza et al., 2020*) with the following exceptions.

1. The maximum $Ca^{2+}$ flux through RyR ($V_{RyR}$) was set to 0. This is the simplest way of removing RyR from the model.
2. A KCa channel, with the same conductance as the basal KCa channel, was introduced into the apical membrane, with the equations for cytosolic and lumenal $[K^+]$ altered to compensate. The apical/basolateral KCa current ratio was varied over a range of values (including 0, which corresponds to the absence of apical KCa), but for most computations the model assumes that the KCa channels have the same density on the apical and basolateral membranes. We note that the KCa current is also determined by the $[Ca^{2+}]$ immediately adjacent to the membrane, and thus the spatial distribution of $Ca^{2+}$ will determine the exact apical/basolateral KCa current ratio. It is only when $[Ca^{2+}]$ is spatially homogeneous that this ratio will equal the apical/basolateral surface area ratio.
3. Na/K-ATPases were introduced into the apical membrane, using the same model and parameters as in *Almássy et al., 2018*. The only change is that the total Na/K-ATPase activity was assumed to be unchanged, with 70% occurring in the basolateral membrane, and 30% in the apical membrane.
4. $V_{PLC}$, which controls the level of agonist stimulation, was set to 0.008 µM/s on the basal membrane, and 0 elsewhere (as in *Pages et al., 2019*; *Vera-Sigüenza et al., 2019*; *Vera-Sigüenza et al., 2020*). However, and in contrast to *Pages et al., 2019*; *Vera-Sigüenza et al., 2019*; *Vera-Sigüenza et al., 2020*, PLC activity was assumed to be independent of $[Ca^{2+}]$. The spatial distribution of PLC has essentially no effect on the model results, and this remains true even when the diffusion coefficient of $IP_3$ is decreased by two orders of magnitude (computations not shown).

The remaining parameters (taken directly from *Vera-Sigüenza et al., 2020*) have been derived from a variety of experimental data over the past decade. For example, the parameters of the IP$_3$R model were determined by fitting to single-channel data (*Cao et al., 2013*; *Cao et al., 2014*; *Siekmann et al., 2012*), while the parameters of many of the ion transporters and exchangers were taken from a variety of published models by other groups, as described in *Palk et al., 2010* and *Vera-Sigüenza et al., 2019*. The parameters of the apical KCa were determined by ensuring qualitative agreement with the experimental data of *Almassy et al., 2012* while, similarly, the anion exchanger parameters were taken directly from the previous modeling work of *Vera-Sigüenza et al., 2018*.

## Acknowledgements

This work was supported by NIH grants RO1DE019245 and RO1DE014756 to DIY, and by a Marsden Fund grant from the Royal Society of New Zealand to JS. AW was supported by F31 DE030670-01A1 from NIH (NIDCR). The authors wish to thank members of the lab for their helpful contributions and discussions during the acquisition of these data.

## Additional information

### Funding

| Funder | Grant reference number | Author |
|---|---|---|
| National Institute of Dental and Craniofacial Research | DE019245 | David I Yule |
| National Institute of Dental and Craniofacial Research | DE014756 | David I Yule |
| Marsden Fund | | James Sneyd |
| National Institute of Dental and Craniofacial Research | F31 DE030670 | Amanda Wahl |

The funders had no role in study design, data collection and interpretation, or the decision to submit the work for publication.

### Author contributions

Takahiro Takano, Formal analysis, Investigation; Amanda M Wahl, Investigation; Kai-Ting Huang, Conceptualization, Methodology; Takanori Narita, Methodology; John Rugis, Software, Visualization, Writing - review and editing; James Sneyd, Conceptualization, Investigation, Methodology, Writing - original draft; David I Yule, Conceptualization, Supervision, Funding acquisition, Writing - original draft, Project administration

### Author ORCIDs

Takanori Narita https://orcid.org/0000-0003-0295-5014
John Rugis http://orcid.org/0000-0002-8009-4152
David I Yule https://orcid.org/0000-0002-6743-0668

### Ethics

Animal experimentation: All animal procedures were approved by University Committee on Animal Resources (UCAR-2001-214E).

### Decision letter and Author response

Decision letter https://doi.org/10.7554/eLife.66170.sa1
Author response https://doi.org/10.7554/eLife.66170.sa2

## Additional files

### Supplementary files
• Transparent reporting form

### Data availability
All data generated or analysed during this study are included in the manuscript and supporting files. Source data files have been provided for Figures 2,3,5,6,7,10,11.Python scripts are available at https://github.com/jrugis/cell_tools (copy archived at https://archive.softwareheritage.org/swh:1:rev:c8fced9a9d8f3b7526a8f8eabfe547c618f2374b).

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
