## [Decision Letter]

**Acceptance summary:**

This paper applies intravital 2-photon microscopy to describe the intracellular calcium signals that drive fluid secretion from salivary gland cells in vivo. Secretion of saliva depends on calcium-dependent chloride and potassium channels, and based on studies of isolated salivary gland cells in vitro, these were previously thought to reside in different regions of the cell and require a propagating calcium wave for their activation. This study reveals a much more localized pattern of calcium signals in the intact organ, providing the first view of how calcium is coupled to secretion in vivo and new insights into how ion channel activity is coordinated to optimize secretion.

**Decision letter after peer review:**

Thank you for submitting your article "The characteristics of intracellular Ca^2+^ signals in vivo necessitate a new model for salivary fluid secretion" for consideration by *eLife*. Your article has been reviewed by 3 peer reviewers, one of whom is a member of our Board of Reviewing Editors, and the evaluation has been overseen Kenton Swartz as the Senior Editor. The following individual involved in review of your submission has agreed to reveal their identity: Peter Thorn (Reviewer #3).

Essential revisions:

The major significance of this work rests on the novel pattern of Ca^2+^ signals associated with secretion in vivo and demonstrating a consequent need for a revised model of apical K channel activation to maintain the driving force for Cl^-^efflux across the apical membrane and fluid secretion.

1. Is a new model needed to describe secretion in vivo? The argument for a new model in which Ca^2+^ activates KCa channels in the apical membrane is based on the observation of low levels of secretion (<5 Hz stimulation) associated with apically localized Ca oscillations. However, these signals are not strictly localized to the apical membrane and would likely activate KCa channels in the basolateral membrane that are close to the tight junctions. In addition, a large increase in secretion is seen for stimuli (5-10 Hz) that propagate Ca to the basal pole, consistent with previous models incorporating activation of basal KCa channels. The contribution of KCa channels in the apical membrane needs to be demonstrated experimentally. This could be addressed by applying blockers to the apical surface, although this may be difficult. Overall, to justify the model, a need for apical KCa channels or evidence of their contribution to secretion, needs to be shown.

2. The apical Ca signals are ascribed solely to release through IP3R, but Orai1 channels, known to be expressed in these cells, are likely to contribute as well. Preferably, Orai1 inhibitors applied systemically (as was done for the cholinesterase inhibitor) should be tested for effects on Ca^2+^ signals and secretion. At the least the contributions of Orai1 channels should be discussed, taking into account previously published work.

3. The computational model does not adequately reproduce the main features of the Ca^2+^ response. It shows a large initial spike upon stimulation (Figure 11B) that is not seen experimentally (Figure 9). It also oscillates at a much lower frequency (<0.2 Hz, similar to in vitro) than measured in vivo (0.5-1 Hz). In addition, the model needs to be tested under more conditions to confirm that it can account for the experimental data; e.g., does it predict the intracellular [Ca^2+^]i gradients, and how increasing stimulation frequency changes the Ca^2+^ response? More data are needed to justify the choice of model parameter values.

4. The absence of local apical Ca^2+^ signals in vitro is not the case for all exocrine acinar cells. The authors should acknowledge that local apical responses have been seen in other acinar cells (e.g. pancreatic acinar cells) in vitro, upon stimulation with physiological concentrations of hormones, low concentrations of neurotransmitter, or directly by IP3.

5. The current data do not allow a fair comparison of the spatiotemporal aspects of Ca^2+^ signaling in vivo and in vitro. While the use of GCaMP6f is a good way to rule out effects of different indicators (e.g., fura-2), other factors are likely to contribute and can be easily tested:

(a) The in vivo experiments were conducted at 37 deg C while the in vitro imaging seems to have been done at room temperature (temperature was not indicated in the Methods). Cooler temperature is likely to affect oscillation frequency as well as the rate of pumping or sequestration which could affect propagation.

(b) The concentrations of ACh used in vitro may not relate to the neural stimulation frequencies in vivo. The 100 nM response in vitro is quite large, so it may not be fair to compare this with low- frequency stimulation in vivo where signals are confined to the apical pole. Perhaps more localized signals would result from lower ACh concentrations.

6. The method for standard deviation (SD) image processing should be described. This approach highlights areas that oscillate, but may de-emphasize areas of tonic elevation which would also be expected to drive secretion. At 5 Hz stimulation it shows that oscillations are confined mostly to the apical region (Figure 7C), but at 10 Hz the cells don't seem to oscillate (Figure 8), so it is not clear why SD images are presented here. It would be more informative to plot a cross-section of GCaMP6 signal from apical to basal before and during stimulation.

*Reviewer #1:*

This work offers a new look at the calcium signals that drive secretion in salivary gland cells. Based on studies of isolated cells in vitro, the common view was that calcium signals originating in the apical pole of the cell propagate to the basal pole, so that chloride channels allow chloride ions to exit across the apical membrane while K channels in the basal membrane are opened to maintain the electrical driving force for chloride exit. Through an elegant use of 2-photon microscopy to monitor the spatial spread of calcium signals in intact glands in vivo, the authors describe a different pattern, in which the signals do not propagate but are much more confined to the apical pole of the cell, and they show that secretion occurs even without significant spread of calcium to the basal pole. These results are significant as they challenge current mechanistic models for secretion, and they imply a functional role for K channels in the apical membrane, which until now have been commonly considered to only function in the basolateral membrane.

However, the paper has several shortcomings that need to be addressed to better support a need for revising the current secretion model. It is not clear whether KCa channels in proximal regions of the basolateral membrane are activated by the apically confined Ca^2+^ and therefore are sufficient to support secretion. A modified computational model intended to support the need for apical KCa channels does not offer strong support, as it does not faithfully reproduce several of the experimentally observed Ca^2+^ responses. Finally, the differences between the spatiotemporal aspects of Ca^2+^ signaling in vivo and in vitro are difficult to assess as critical conditions such as temperature and agonist concentration were not matched in the two experimental scenarios.

*Reviewer #2:*

In this study Dr. Yule and his colleagues describe very impressive and painstaking work in which they have monitored for the first time Ca^2+^ signals and salivary secretion in vivo in a mouse model which expressed the Ca^2+^ indicator protein, GCamp6F, in salivary glands. Previously reported studies measured fluid secretion in vivo by injecting the animals with pilocarpine and collecting the oral secretion or by using protocols where the gland is perfused. However, cytosolic [Ca^2+^] changes, which are correlated with, and required for, salivary fluid secretion have been predominantly measured in isolated acinar or cell cluster preparations and some in tissue slices. By using intravital multiphoton (MP) microscopy together with stimulation of the neural input to the gland, the authors were able to measure the spatiotemporal properties of Ca^2+^ signals within the acinar cells and collect saliva that is secreted into the oral cavity. Thus, this study is an important breakthrough in this field as it reveals the exact [Ca^2+^] changes associated with salivary gland fluid secretion.

This study shows that major spatiotemporal properties of the Ca^2+^ signals detected in vivo are markedly different from those previously reported using isolated acinar cells (by this group and others) which demonstrated that Ca^2+^ signals are initiated apically but propagate as a wave globally across the cell without generation of Ca^2+^ gradients. The present findings show that following moderate stimulation, apically localized oscillatory Ca^2+^ signals are generated that do not propagate globally and cause minimal secretion. However, at stimulation intensities optimum for secretion, there is a spread of Ca^2+^ across the cell although with an apical-basal standing Ca^2+^ gradient. Based on these differences the authors have proposed a revised model which suggests that the apical region contains all the machinery needed for saliva secretion and that Ca^2+^ released predominately in the apical region, via IP3R, can regulate the function of locally situated ion channels without requirement for a propagated wave of increased [Ca^2+^] across the cell. As I have noted above the data described in this manuscript provide new understanding of the physiological response of salivary gland to neuronal stimulation and how fine-tuning of the spatiotemporal properties of the Ca^2+^ signals regulate fluid secretion.

The main question I have is regarding stimulus strengths that induce optimal levels of secretion (10 and 30Hz) where there is propagation of Ca^2+^ across the cell and almost 3-fold higher secretion relative to that at 5Hz. While increase in % responding cells at higher stimulus strengths could result in higher total secretion, possible contribution of Ca^2+^ influx pathways in the spread of Ca^2+^ in individual acinar cells, e.g. via Orai1 that is localized in the apical region of the cell, cannot be ruled out. Also not clear is whether the spread enhances secretion. Thus, when stimulus is low, secretion is minimal and supported only by release and oscillatory apical [Ca^2+^]. At higher stimulus strengths there might be an additional Ca^2+^ influx component which is activated very early after stimulation, likely during IP3-mediated Ca^2+^ release. Since the basolateral membrane extends all the way to the tight junction, is it possible that several components shown basally localized in the model could actually be quite close to the apical membrane region, near the tight junctions.

Thus, it is important to distinguish between initial stimulation of secretion vs regulation of sustained secretion. For example; the last sentence in the abstract clearly, and correctly, states "salivary secretion can be efficiently stimulated by apically localized Ca^2+^ signals".

*Reviewer #3:*

This paper describes a nice set of experiments where the authors have measured calcium responses in salivary acinar cells from living animals and, importantly, correlated these with measures of salivary secretion. The work is very well performed and the data interesting and definitely an advance in the field. I cannot think of another study that combines, so well, cellular calcium responses with genuine physiological output.

Strengths

Intravital images are very good and the combination with fluid secretion gives a nice correlation between calcium responses and physiological output – something which is impossible in vitro.

Good initial analysis of the images and convincing data where the calcium responses are further analysed.

The conclusions that local calcium responses can drive physiological responses is interesting in the context of salivary glands.

Weaknesses

The authors claim that local responses are not seen in vitro is dependent on taking a very narrow view of exocrine cells. Local responses are readily seen in other exocrine cells, like pancreatic acinar cells. These local responses can be driven directly by IP3, low concentrations of neurotransmitter, or physiological concentrations of hormones. Furthermore, these responses drive enzyme secretion. Therefore, although these local responses have not been measured intravitally, it is highly likely that they are the native response in the pancreas.

The new model put forward is conjecture. It is built entirely around the observations of localised calcium responses but has no supporting data on changes in membrane potential or activation of specific channels.

The in vitro GCaMP data is not convincing. GCaMP and Fura have different affinities for calcium and so this is a serious potential confounder in their experiments. For the in vitro work, the isolated cells are clearly fragmented and the cell chosen is at the edge of the cell cluster. It would be expected that the cells will completely lose their polarisation and structural factors that might limit the calcium response.

While I like the analysis, I think that much of it is unnecessary. The authors, for example, show that regions that respond well at low stimulation also respond well at high levels of stimulation. That's fine, but the authors then do not explore what the basis of these differences are.

[Editors' note: further revisions were suggested prior to acceptance, as described below.]

Thank you for resubmitting your work entitled "The characteristics of intracellular Ca^2+^ signals in vivo necessitate a new model for salivary fluid secretion" for further consideration by *eLife*. Your revised article has been evaluated by Kenton Swartz (Senior Editor) and a Reviewing Editor.

Thank you for the revised manuscript with the additional data and clarifications. Your responses have addressed most of the major concerns of the reviewers. The only remaining issue is whether KCa channels in the apical membrane are required for secretion. I think this is central to the significance of the paper. The study shows beautifully how Ca signals are mostly apical and do not propagate to the basal membrane in vivo, but the most interesting implication is that it necessitates having KCa channels near or in the apical membrane. To broaden the appeal of your paper to more than exocrine gland physiologists it will be important to establish this point clearly.

We understand that an experimental approach (luminal application of blockers) to establish the role of apical KCa channels is not feasible. Thus, the model offers the best opportunity to test this idea. The model needs a fuller description, not including all the details and equations, but rather a description of the salient features that will allow the reader to grasp how it works, and what assumptions it makes. The comments below summarize the uncertainties in the model's operation that need to be addressed:

1. How are apical, cytoplasmic, and basal regions defined in the model, and how do they correspond to the same regions in the experimental data? Throughout the paper there are references to "apical PM" or "apical region." The apical PM is the membrane bounded by tight junctions, but what does "apical region" mean – does this include lateral membrane between the apical PM and the nucleus? The model cartoon in Figure 13 does not explain the distinction between apical and basolateral. Likewise, on p. 12: "We emphasise that, in the model, KCa channels and Na/K- ATPases are also present in the basal membrane, and thus the secretory machinery is not restricted to the apical membrane." Is "basal" here referring to lateral membrane as well? I imagine that some of this vagueness is because the anatomy is complex and the localization of channels is not precisely known, but this needs to be clarified, particularly for non-expert readers.

2. How is the KCa current simulated? This is a crucial part of the model, but it is not clear how this was done. The only description is (p. 25) "The apical/basal K^+^ current ratio was determined by the apical/basal surface area ratio." After Ca^2+^ is released from the ER, is it assumed to diffuse to the lateral membrane where it activates KCa channels? (lateral membrane is defined here as basolateral membrane between the tight junction and the nucleus, corresponding to the "cytoplasmic" region in the experimental Ca measurements.) Or is only the basal Ca^2+^ signal being used to determine KCa current? The data in Figure 10 show that there is significant Ca^2+^ elevation in the cytoplasmic region below the apical membrane even at low stimulation frequencies, which could activate channels in the lateral membrane. What spatial distribution of KCa channels is assumed in the model? These things need to be clearly and simply explained.

3. The model shows that with no apical KCa channels (but the same total number) you still get ~80% of the maximal response (Figure 14F). This is almost maximal and implies only a modest effect of true apical PM channels. Figure 14F Y axis should be plotted from 0 rather than 60, in order to give a truer impression of the effect of KCa channel localization. There is also some confusion here: Figure 14B shows only 50% of max response if you remove apical KCa channels – but I assume that this is because the total number of channels was not constant in this case, resulting in fewer KCa channels than in Figure 14F. If so, that is not a fair comparison.

4. On p. 17 there is a reference to Supp. Figure 4, but that figure was not included.

5. I would suggest some rewording of the title, as it seems too focused on what was done, rather than the most significant result. Perhaps you can revise it to focus on localized calcium signaling that engages apical K channels in vivo.

---

## [Author Response]

Essential revisions:The major significance of this work rests on the novel pattern of Ca^2+^ signals associated with secretion in vivo and demonstrating a consequent need for a revised model of apical K channel activation to maintain the driving force for Cl^-^efflux across the apical membrane and fluid secretion.1. Is a new model needed to describe secretion in vivo? The argument for a new model in which Ca^2+^ activates KCa channels in the apical membrane is based on the observation of low levels of secretion (<5 Hz stimulation) associated with apically localized Ca oscillations. However, these signals are not strictly localized to the apical membrane and would likely activate KCa channels in the basolateral membrane that are close to the tight junctions. In addition, a large increase in secretion is seen for stimuli (5-10 Hz) that propagate Ca to the basal pole, consistent with previous models incorporating activation of basal KCa channels. The contribution of KCa channels in the apical membrane needs to be demonstrated experimentally. This could be addressed by applying blockers to the apical surface, although this may be difficult. Overall, to justify the model, a need for apical KCa channels or evidence of their contribution to secretion, needs to be shown.

While a direct demonstration that apical KCa channels participate in the secretory mechanism would be ideal, we believe that the experiments proposed (local application of blockers to the apical face of acini) is not practically feasible. Specifically, given the architecture of an acinus, it is not possible to effectively isolate the apical plasma membrane (~1-5 % of PM in the center of a group of cells) to apply a pharmacological agent locally with any degree of certainty without it acting basolaterally. In addition, it is difficult to imagine how this could be performed in every apical domain in the gland to assess its effect on fluid secretion into the oral cavity following neural stimulation. Further, this problem is not obviously amenable to any straightforward genetic approach. Our previous data has demonstrated the presence of apical KCa channels in salivary gland acinar cells and we provide a more complete discussion of evidence for their functional importance.

2. The apical Ca signals are ascribed solely to release through IP3R, but Orai1 channels, known to be expressed in these cells, are likely to contribute as well. Preferably, Orai1 inhibitors applied systemically (as was done for the cholinesterase inhibitor) should be tested for effects on Ca^2+^ signals and secretion. At the least the contributions of Orai1 channels should be discussed, taking into account previously published work.

We have performed new in vivo experiments using the Orai channel blocker, GSK7975A, which show that the sustained phase of the Ca^2+^ signal following extended stimulation with physiological stimulus strengths is blunted by the Orai antagonist. The initial peak is largely unaffected, consistent with the Ca^2+^ response to sustained stimulation being mediated by Ca^2+^ release and subsequent Ca^2+^ influx.

3. The computational model does not adequately reproduce the main features of the Ca^2+^ response. It shows a large initial spike upon stimulation (Figure 11B) that is not seen experimentally (Figure 9). It also oscillates at a much lower frequency (<0.2 Hz, similar to in vitro) than measured in vivo (0.5-1 Hz). In addition, the model needs to be tested under more conditions to confirm that it can account for the experimental data; e.g., does it predict the intracellular [Ca^2+^]i gradients, and how increasing stimulation frequency changes the Ca^2+^ response? More data are needed to justify the choice of model parameter values.

The model has been modified and the computational aspects of these studies have been expanded to test more conditions. The modified model now also more faithfully reflects the characteristics of the Ca^2+^ signals observed experimentally. A more detailed discussion of the model and predictions is also now provided, specifically to explicitly state that basolateral KCa channels in the original (and present) model are present throughout the PM to regions directly adjacent to tight junctions. We now test the model responses under more conditions. We show that increasing stimulation increases the frequency of oscillations, and decreases the latency (new Figure 14A), in agreement with data. We also compare model and experimental line scans and show that the [Ca^2+^] gradient through the cell in the model is qualitatively similar to that observed experimentally. We test model behaviour for lower values of the IP_3_ diffusion coefficient, and show that the behaviour remains unchanged (results not shown, as they are so similar to the results obtained with a larger IP_3_ diffusion coefficient. However, these results can be included as supplementary data should this be required). We simulated the model with periodic stimulation of PLC activity, to mimic the effects of periodic neural stimulation, and showed that the [Ca^2+^] traces and secretion remain essentially unchanged (although there are some minor quantitative differences). Again, because of the similarity with the results already shown here we do not include these results explicitly, we just state the result. We now give a more detailed discussion of where the model parameter values are derived. Since these parameter values have been developed over many years, and have already been presented in multiple papers, we do not give full details again in the present submission. For example, the parameters that govern our model of the IPR were determined by fitting the model to single-channel data, and this work is described in a series of papers between 2009 and 2019 (Refs 1-7 below). However, any interested reader is now told more precisely where further details about parameter values can be found.

1. Gin, E., L.E. Wagner, D.I. Yule, and J. Sneyd, Inositol trisphosphate receptor and ion channel models based on single-channel data. Chaos, 2009. 19(3): p. 037104.

2. Gin, E., M. Falcke, L.E. Wagner, 2nd, D.I. Yule, and J. Sneyd, A kinetic model of the inositol trisphosphate receptor based on single-channel data. Biophys J, 2009. 96(10): p. 4053-62.

3. Gin, E., M. Falcke, L.E. Wagner, D.I. Yule, and J. Sneyd, Markov chain Monte Carlo fitting of single-channel data from inositol trisphosphate receptors. J Theor Biol, 2009. 257(3): p. 460-74.

4. Siekmann, I., L.E. Wagner, 2nd, D. Yule, E.J. Crampin, and J. Sneyd, A kinetic model for type I and II IP3R accounting for mode changes. Biophys J, 2012. 103(4): p. 658-68.

5. Siekmann, I., L.E. Wagner, D. Yule, E.J. Crampin, and J. Sneyd, A Park/Drive Model for the Inositol-Trisphosphate Receptor (IPR). Biophysical Journal, 2012. 102(3): p. 110a-110a.

6. Cao, P., X. Tan, G. Donovan, M.J. Sanderson, and J. Sneyd, A deterministic model predicts the properties of stochastic calcium oscillations in airway smooth muscle cells. PLoS Comput Biol, 2014. 10(8): p. e1003783.

7. Cao, P., G. Donovan, M. Falcke, and J. Sneyd, A stochastic model of calcium puffs based on single-channel data. Biophys J, 2013. 105(5): p. 1133-42.

4. The absence of local apical Ca^2+^ signals in vitro is not the case for all exocrine acinar cells. The authors should acknowledge that local apical responses have been seen in other acinar cells (e.g. pancreatic acinar cells) in vitro, upon stimulation with physiological concentrations of hormones, low concentrations of neurotransmitter, or directly by IP3.

A discussion of the numerous reports of apically localized Ca^2+^ signals in acutely isolated pancreatic acinar cells is now included.

5. The current data do not allow a fair comparison of the spatiotemporal aspects of Ca^2+^ signaling in vivo and in vitro. While the use of GCaMP6f is a good way to rule out effects of different indicators (e.g., fura-2), other factors are likely to contribute and can be easily tested:(a) The in vivo experiments were conducted at 37 deg C while the in vitro imaging seems to have been done at room temperature (temperature was not indicated in the Methods). Cooler temperature is likely to affect oscillation frequency as well as the rate of pumping or sequestration which could affect propagation.(b) The concentrations of ACh used in vitro may not relate to the neural stimulation frequencies in vivo. The 100 nM response in vitro is quite large, so it may not be fair to compare this with low- frequency stimulation in vivo where signals are confined to the apical pole. Perhaps more localized signals would result from lower ACh concentrations.

We have now provided new in vitro data that demonstrates that acini are more sensitive to agonist stimulation and oscillations are more rapid at 37^o^C, but the Ca^2+^ are still invariably global at physiological temperature.

6. The method for standard deviation (SD) image processing should be described. This approach highlights areas that oscillate, but may de-emphasize areas of tonic elevation which would also be expected to drive secretion. At 5 Hz stimulation it shows that oscillations are confined mostly to the apical region (Figure 7C), but at 10 Hz the cells don't seem to oscillate (Figure 8), so it is not clear why SD images are presented here. It would be more informative to plot a cross-section of GCaMP6 signal from apical to basal before and during stimulation.

We describe the method for SD image generation. While we contend that this method is an effective means of visualizing the sub-cellular gradients (“average” images generated from the time series are essentially indistinguishable from the SD image and highlight apical-basal gradients), we now provide plots of multiple ROI across the acinus to better visualize the gradients (Figures 8/9). We also provide a new figure where line scans have been performed across an acinus (figure 10). Both these approaches demonstrate that the magnitude of the basal Ca^2+^ signal is ~20-30% of that in the apical domain.

Reviewer #1:[…] The paper has several shortcomings that need to be addressed to better support a need for revising the current secretion model. It is not clear whether KCa channels in proximal regions of the basolateral membrane are activated by the apically confined Ca^2+^ and therefore are sufficient to support secretion.

We appreciate the opportunity to clarify this point. In the model the basolateral membrane extends all the way to the tight junction, and thus a fraction of the basal KCa channels are close enough to the apical membrane to be activated by Ca^2+^. This fact has been explicitly stated in the new version of the manuscript. The effect of this localization can be seen in the blue curve in new Figure 14B, which shows how, in the old model modified to have no propagated apical-to-basal Ca^2+^ wave, there is still an increase in secretion upon stimulation. However, this increase is only about 50% of the saliva secretion attained if apical KCa channels are included, as in the new version of the model (new Figure 14B, red curve).

A modified computational model intended to support the need for apical KCa channels does not offer strong support, as it does not faithfully reproduce several of the experimentally observed Ca^2+^ responses.

The large initial peak in the first version of the model has now been corrected by making a slight change to the way in which we model the activity of PLC. In the first version of the model PLC activity was a function of [Ca^2+^]. In the new version of the model this dependence on [Ca^2+^] has been removed (as described on new page 24). This change has removed the artifact of the large initial spike, but has made no difference to the conclusions drawn from the model results. The model simulations now closely resemble the data, in that they are oscillations (with the correct frequency of around 0.5 to 1 Hz at high stimulation) superimposed on an increasing baseline as seen in new figure 14 A and discussed on page 12.

Finally, the differences between the spatiotemporal aspects of Ca^2+^ signaling in vivo and in vitro are difficult to assess as critical conditions such as temperature and agonist concentration were not matched in the two experimental scenarios.

We agree with the reviewer and have now included data which investigates the characteristics of the Ca^2+^ signals at 37^o^C in isolated GCamp6F expressing acinar cells. We now show that the individual cells are more sensitive to secretagogue stimulation and exhibit more rapid oscillations at 37^o^C. Nevertheless, at all concentrations of secretagogues, the signals originate in the apical aspects of the cell and invariably globalize and thus the indicator or experimental temperature cannot explain the in vivo data.

Reviewer #2:[…] The main question I have is regarding stimulus strengths that induce optimal levels of secretion (10 and 30Hz) where there is propagation of Ca^2+^ across the cell and almost 3-fold higher secretion relative to that at 5Hz. While increase in % responding cells at higher stimulus strengths could result in higher total secretion, possible contribution of Ca^2+^ influx pathways in the spread of Ca^2+^ in individual acinar cells, e.g. via Orai1 that is localized in the apical region of the cell, cannot be ruled out.

This is an important point because it is highly likely that Ca^2+^ influx pathways play a major role in maintaining the Ca^2+^ signal, and thus secretion, especially during prolonged stimulation. We now include data which shows that systemic administration of an Orai channel blocker GSK7579A markedly inhibits in vivo Ca^2+^ signals to physiologically optimum nervous stimuli. Notably, but only after extended stimulation (~5 min) and has little impact on the initial phase of the Ca^2+^ rise. These data are included as new figure 4.

Also not clear is whether the spread enhances secretion. Thus, when stimulus is low, secretion is minimal and supported only by release and oscillatory apical [Ca^2+^]. At higher stimulus strengths there might be an additional Ca^2+^ influx component which is activated very early after stimulation, likely during IP3-mediated Ca^2+^ release. Since the basolateral membrane extends all the way to the tight junction, is it possible that several components shown basally localized in the model could actually be quite close to the apical membrane region, near the tight junctions.Thus, it is important to distinguish between initial stimulation of secretion vs regulation of sustained secretion. For example; the last sentence in the abstract clearly, and correctly, states "salivary secretion can be efficiently stimulated by apically localized Ca^2+^ signals".

We obviously accept this idea and as stated in response to the Reviewer 1, our model originally incorporated KCa channels in the basolateral membrane immediately abutting the tight-junctions and incorporates the evidence that some will be activated by the reduced Ca^2+^ signal seen in the basal region. Secretion predictions from the model varying the ratio of apical:basal KCa are now included and support the idea that secretion is optimum with a proportion of apical KCa, activated by the large increase in Ca^2+^ which occur in the extreme apical domain.

Reviewer #3:[…] The authors claim that local responses are not seen in vitro is dependent on taking a very narrow view of exocrine cells. Local responses are readily seen in other exocrine cells, like pancreatic acinar cells. These local responses can be driven directly by IP3, low concentrations of neurotransmitter, or physiological concentrations of hormones. Furthermore, these responses drive enzyme secretion. Therefore, although these local responses have not been measured intravitally, it is highly likely that they are the native response in the pancreas.

The Authors acknowledge that there is a rich literature in isolated pancreatic acinar cells demonstrating that local Ca^2+^ signals occur at low agonist concentrations and exposure to IP_3_. Additional references have been added to the text to reflect this. A discussion point is that the majority of these studies were performed in whole-cell patch clamped isolated cells, where cytoplasmic buffering is imposed by the internal pipette solution and which itself could influence the spatial characteristics of the Ca^2+^ signals and thus it would still be informative to observe Ca^2+^ signals stimulated in vivo in the exocrine pancreas.

The new model put forward is conjecture. It is built entirely around the observations of localised calcium responses but has no supporting data on changes in membrane potential or activation of specific channels.

The model has been developed over a number of years and is faithfully built on experimental data. More details are given regarding the model parameters in the revised paper, which have been derived experimentally and published over many years. While it is correct that the present study did not monitor any effectors of the local Ca^2+^ signal in vivo, a previous study (Almassy et al., JGP 139. 121-133. 2012) exhaustively demonstrated that there were apical KCa3.1 and KCa1.1 in the apical PM of salivary acinar cells, which were activated by local uncaging of Ca^2+^. A more extensive discussion of this previous data has been added to the text in the revised manuscript.

The in vitro GCaMP data is not convincing. GCaMP and Fura have different affinities for calcium and so this is a serious potential confounder in their experiments. For the in vitro work, the isolated cells are clearly fragmented and the cell chosen is at the edge of the cell cluster. It would be expected that the cells will completely lose their polarisation and structural factors that might limit the calcium response.

We agree that a more thorough characterization of the attributes of the Ca^2+^ signal in vitro would strengthen our case that there is a real difference between in vivo and isolated cells. However, we stress that in 20 years working on Ca^2+^ signaling in salivary gland acinar cells, stimulated by secretagogue or IP_3_ directly, we have never observed an apical confined Ca^2+^ rise. We now include additional data, from experiments performed at 37^o^C and over a wider range of secretagogue concentrations.

While I like the analysis, I think that much of it is unnecessary. The authors, for example, show that regions that respond well at low stimulation also respond well at high levels of stimulation. That's fine, but the authors then do not explore what the basis of these differences are.

The Authors contend that this analysis establishing the “baseline” characteristics of in vivo Ca^2+^ signals in salivary acinar cells for the first time is important *per se*. The response heterogeneity observed is a feature which can only really be meaningful when studied in vivo, given that any variability in response in isolated cells could be due to the isolation procedure. Exploring further the underlying mechanisms will be the subject of future study.

[Editors' note: further revisions were suggested prior to acceptance, as described below.]

Thank you for the revised manuscript with the additional data and clarifications. Your responses have addressed most of the major concerns of the reviewers. The only remaining issue is whether KCa channels in the apical membrane are required for secretion. I think this is central to the significance of the paper. The study shows beautifully how Ca signals are mostly apical and do not propagate to the basal membrane in vivo, but the most interesting implication is that it necessitates having KCa channels near or in the apical membrane. To broaden the appeal of your paper to more than exocrine gland physiologists it will be important to establish this point clearly.We understand that an experimental approach (luminal application of blockers) to establish the role of apical KCa channels is not feasible. Thus, the model offers the best opportunity to test this idea. The model needs a fuller description, not including all the details and equations, but rather a description of the salient features that will allow the reader to grasp how it works, and what assumptions it makes. The comments below summarize the uncertainties in the model's operation that need to be addressed:

The Authors wish to thank the Reviewers and Editor for their constructive comments. In response we have made changes to the text and modified figures. We believe this has further clarified our description of the architecture of the acini as it relates to the experimental and computational approaches with the overarching goal of making the paper more accessible to a wider readership. Specifically, this includes a more extensive description of the model, including an additional supplemental figure and additional results of the model simulations. We provide a point-by-point response to the comments provided below.

1. How are apical, cytoplasmic, and basal regions defined in the model, and how do they correspond to the same regions in the experimental data? Throughout the paper there are references to "apical PM" or "apical region." The apical PM is the membrane bounded by tight junctions, but what does "apical region" mean – does this include lateral membrane between the apical PM and the nucleus? The model cartoon in Figure 13 does not explain the distinction between apical and basolateral. Likewise, on p. 12: "We emphasise that, in the model, KCa channels and Na/K- ATPases are also present in the basal membrane, and thus the secretory machinery is not restricted to the apical membrane." Is "basal" here referring to lateral membrane as well? I imagine that some of this vagueness is because the anatomy is complex and the localization of channels is not precisely known, but this needs to be clarified, particularly for non-expert readers.

In the introduction, we now provide a paragraph describing the general architecture of exocrine acinar cells, including making it clear that basal and lateral membranes are functionally identical. The cartoon in Figure 13 has been amended to illustrate this architecture. In an effort to define the “apical region” we also highlight published data demonstrating that the apical ER and apical plasma membrane are intimately associated and can be considered a region of privileged communication. In the experimental figures, the apical domain is defined as in the line scan data, where the apical domain is considered to be 3 μm from the apical PM. The line scan and positioning of ROIs in experimental data is designed primarily to illustrate the spatial heterogeneity of the Ca^2+^ signal. It is possible this volume encompasses some lateral membrane in the region defined as “cytoplasmic”, however this does not impact the secretion model because of how this Ca^2+^ release is modelled (see below).

In the model the apical, basal and lateral membranes of the model are clearly defined by anatomical reconstructions from Z stacks in which the TMEM channels and the Na/K ATPases are labelled. This procedure is discussed in detail in previous papers, but here, for clarity, we now include a three-dimensional plot of the cell (supplemental figure 5) on which all the computations in this paper were performed. The acinar membranes are labelled in red and the basolateral membranes in blue. Thus, the reader can now obtain a much clearer intuitive understanding of the model domain, without having to read our previous papers. Although there is no such clear-cut experimentally-based definition of apical or basal “regions”, this is unimportant for the model, in which the concept of apical and basal “regions” plays essentially no role. This is because the IPR flux is modelled as a boundary flux through the apical membrane, while PLC activity is restricted to the basal membrane. Thus, the spatial positioning of these two critical Ca pathways is determined directly (and reasonably precisely) by experimental data. Of course, this requires some simplifying assumptions in the model; for example, the IPR Ca^2+^ flux is not exactly a boundary flux, but comes through IPR which we know are situated very close (within approximately 50 nm) of the TMEM channels, which are certainly on the apical membrane. However, this simplifying assumption appears to introduce no significant quantitative differences; we know this because we have also run simulations in which the IPR Ca^2+^ flux occurs in an apical “region” consisting of all the grid points within 3 (or 2, or 1, or 0.5) microns of the apical membrane, and the model simulations show no significant quantitative differences in any of these cases. We now explain this in more detail on new pages 24 and 25.

Although the model does technically make a distinction between the basal membranes and the lateral membranes (i.e., the lateral surface triangles are labelled differently, can be colored differently, and are constructed so as to be conformal with the lateral surface triangles of the immediately adjacent cell) this distinction is of no importance in

the current simulations, which (without exception) assume that all the ion channels and transporters (with the exception of TMEM, which is restricted to the apical membrane) are uniformly distributed across both the basal and lateral membranes. This means that Ca^2+^ released from the apical “region” (which, in the model, means Ca^2+^ influx across the surface triangles corresponding to the apical membrane) can diffuse to neighboring basolateral surface triangles and activate KCa channels on the basolateral surface. This is why the model exhibits substantial fluid flow even when there are no KCa channels in the apical membrane.

2. How is the KCa current simulated? This is a crucial part of the model, but it is not clear how this was done. The only description is (p. 25) "The apical/basal K^+^ current ratio was determined by the apical/basal surface area ratio." After Ca^2+^ is released from the ER, is it assumed to diffuse to the lateral membrane where it activates KCa channels? (lateral membrane is defined here as basolateral membrane between the tight junction and the nucleus, corresponding to the "cytoplasmic" region in the experimental Ca measurements.) Or is only the basal Ca^2+^ signal being used to determine KCa current? The data in Figure 10 show that there is significant Ca^2+^ elevation in the cytoplasmic region below the apical membrane even at low stimulation frequencies, which could activate channels in the lateral membrane. What spatial distribution of KCa channels is assumed in the model? These things need to be clearly and simply explained.

We now explain in much more detail (new page 25) exactly how activation of the KCa channels works in the model. Briefly (and as we describe above also), the channels on each surface triangle are affected only by the average [Ca^2+^] in the tetrahedron that contains that triangle. Thus Ca^2+^ released from the apical region can diffuse a short distance to the neighboring basolateral membrane to activate KCa channels there.

3. The model shows that with no apical KCa channels (but the same total number) you still get ~80% of the maximal response (Figure 14F). This is almost maximal and implies only a modest effect of true apical PM channels. Figure 14F Y axis should be plotted from 0 rather than 60, in order to give a truer impression of the effect of KCa channel localization. There is also some confusion here: Figure 14B shows only 50% of max response if you remove apical KCa channels – but I assume that this is because the total number of channels was not constant in this case, resulting in fewer KCa channels than in Figure 14F. If so, that is not a fair comparison.

In the previous resubmission we acknowledged that the model predicts that the absence of apical KCa channels reduces salivary secretion by ~25%. We contend that this is not insignificant. Nevertheless, there is indeed only a partial effect from the presence of apical KCa channels, as the model is capable of significant fluid secretion in their absence. We now emphasize more strongly throughout that the model predicts that apical KCa channels, although not *necessary* for fluid flow, appear to be required for *optimal* fluid flow, hence the change in the title for the manuscript. We have modified the figure as requested by the reviewer and added simulations for additional levels of stimulus strengths.

We also now explain at greater length (and, we hope, more clearly) that the curve with no apical KCa in Figure 14B corresponds to simulations from the *old* model, not the new one, which explains the quantitative differences with Figure 14F. The curves in Figure 14 B are designed to show how the old model is poor at generating fluid flow in the absence of apical KCa, and thus a new model is needed. Of course, we note that even the old model can generate *some* fluid flow even in the absence of apical KCa. In both models it is simply a matter of how much fluid flow there is, not whether fluid flow exists at all.

4. On p. 17 there is a reference to Supp. Figure 4, but that figure was not included.

Supp Figure 4- (Figure 5-Supp Figure 4) is now included.

5. I would suggest some rewording of the title, as it seems too focused on what was done, rather than the most significant result. Perhaps you can revise it to focus on localized calcium signaling that engages apical K channels in vivo.

We have amended the title to “Highly Localized intracellular Ca^2+^ signals promote optimal salivary gland fluid secretion”, which we feel better reflects the data.